
**Differences of the inverted terrestrial ecosystem carbon flux between using GO-**
**SAT and OCO-2 XCO₂ retrievals**
Hengmao Wang[1], Fei Jiang[1,2*], Jun Wang[1], Weimin Ju[1], Jing M. Chen[1,3]
*1 Jiangsu Provincial Key Laboratory of Geographic Information Science and Technology, International Institute for*
*Earth System Science, Nanjing University, Nanjing, 210023, China*
*2 Jiangsu Center for Collaborative Innovation in Geographical Information Resource Development and Application,*
*Nanjing, 210023, China*
*3, Department of Geography, University of Toronto, Toronto, Ontario M5S3G3, Canada*
**Abstract**
In this study, both the Greenhouse Gases Observing Satellite (GOSAT) and the Orbiting Car-
bon Observatory 2 (OCO-2) XCO₂ retrievals are assimilated within the GEOS-Chem 4D-Var assim-
ilation framework to constrain the terrestrial ecosystem carbon flux during Jul 1, 2014 to Dec 31,
2015. The inverted global and regional carbon fluxes during Jan 1 to Dec 31, 2015 are shown and
discussed. Surface CO₂ mixing ratios from 47 surface flask sites and XCO₂ measurements from 13
TCCON sites are used to evaluate the simulated concentrations with the posteriori carbon fluxes. The
results show that globally, the terrestrial ecosystem carbon sink (excluding biomass burning emis-
sions) estimated from GOSAT data is stronger than that inferred from OCO-2 data, and the annual
atmospheric CO₂ growth rate estimated from GOSAT data is more consistent with the estimate of
GCP 2017. Regionally, in most regions, the land sinks inferred from GOSAT data are also stronger
than those from OCO-2 data. Compared with the prior fluxes, the carbon fluxes in northern temperate
regions change most, followed by tropical and southern temperate regions, and the smallest changes
occur in boreal regions. Basically, in temperate regions, the inferred land sinks are significantly in-
creased, while those in tropical regions are decreased. The different changes in different regions are
mainly related to the spatial coverage and the amount of XCO₂ data in these regions. Compared with
CT2016, the inferred carbon sinks are comparable in most temperate regions, but much weaker in

---

* Corresponding author: Tel.: +86-25-83597077; Fax: +86-25-83592288; E-mail address: jiangf@nju.edu.cn





boreal and tropical regions. Evaluations using flask and TCCON observations suggest that GOSAT
and OCO-2 data, can effectively improve the carbon flux estimates in the northern hemisphere, while
in the southern hemisphere the optimized carbon sinks may be overestimated, especially for GOSAT
data.
**Keywords**: Terrestrial ecosystem carbon flux, inversion, GOSAT, OCO-2, GEOS-Chem
**1. Introduction**
Atmospheric inverse modeling is an effective method for quantifying surface carbon fluxes at
global and regional scales using the gradient of $CO_2$ measurements. Inversion studies based on in-
situ $CO_2$ observations agree well on global carbon budget estimates but differ greatly on regional
carbon flux estimates and the partitioning of land and ocean fluxes as well,  mainly due to the sparse-
ness of observations in tropics, southern hemisphere oceans and the majority of continental interiors
such as those in South America, Africa, and Boreal Asia (Peylin el al., 2013). Satellite observations
offer an attractive means to constrain atmospheric inversions with their extensive spatial coverage
over remote regions. Studies have shown that satellite observations, though with lower precision than
in-situ measurements, can improve the carbon flux estimates (Rayner and O Brien, 2001; Pak and
Prather, 2001; Houweling et al., 2004; Baker et al., 2006; Chevallier., 2007; Miller et al., 2007; Kady-
grov et al., 2009; Hungershoefer et al., 2010).
Satellite sensors designed specifically to measure atmospheric $CO_2$ concentrations, have been
in operation in recent years. The Greenhouse Gases Observing Satellite (GOSAT) (Kuze et al.,
2009), being the first satellite mission dedicated to observing $CO_2$ from space, was launched in
2009. The National Aeronautics and Space Administration (NASA) launched the Orbiting Carbon
Observatory 2 (OCO-2) satellite in 2014 (Crisp et al., 2017; Eldering et al., 2017). China's first $CO_2$
monitoring satellite (TanSat) was launched in 2016 (Wang et al, 2017; Yang et al, 2017). These sat-
ellites measure near-infrared sunlight reflected from the surface in $CO_2$ spectral bands and the $O_2$





A-band to retrieve column-averaged dry-air mole fractions of $CO_2$ (XCO2), aiming to improving the
estimation of spatial and temporal distributions of carbon sinks and sources. A number of inversions
have utilized GOSAT XCO2 retrievals to infer surface carbon fluxes (Takagi et al., 2011; Basu et
al., 2013; Maksyutov et al., 2013; Saeki et al., 2013; Chevallier et al., 2014; Deng et al., 2014;
Houweling et al., 2015; Deng et al, 2016). Although large uncertainty reductions were achieved
for regions which are under-sampled by in-situ observations, these studies didn't give robust re-
gional carbon flux estimates. There are large spreads in regional flux estimates in some regions
among these inversions. Furthermore, regional flux distributions inferred from GOSAT XCO2 data
are significantly different from those inferred from in-situ observations. For instance, several stud-
ies using GOSAT retrievals reported a larger than expected carbon sink in Europe (Basu et al.,
2013; Chevallier et al., 2014; Deng et al., 2014; Houweling et al., 2015). The validity of this large
Europe carbon sink derived from GOSAT retrievals is in intense debate and efforts to achieve a
consensus estimate of Europe carbon sinks are still ongoing (Reuter et al., 2014; Feng et al., 2016;
Reuter et al., 2017).
Compared with GOSAT, OCO-2 has a higher sensitivity near the surface, much finer footprints
and more extended spatial coverage, and thus has the potential to better constrain the surface carbon
flux inversion (Eldering et al., 2017). Studies have used OCO-2 XCO2 data to estimate carbon flux
anomalies during recent El Nino events (Chatterjee et al., 2017; Patra et al., 2017; Heymann et al.,
2017; Liu et al., 2017). Nassar et al. (2017) applied OCO-2 XCO2 data to infer emissions from large
power plants. Miller et al. (2018) evaluated the potential of OCO-2 XCO2 data in constraining re-
gional biospheric $CO_2$ fluxes and found that in the current state of development, OCO-2 observa-
tions can only provide a reliable constraint on $CO_2$ budget at continental and hemispheric scales. At
present, it is still not clear whether with the improved monitoring capabilities, current OCO-2 ob-
servations have a greater potential than GOSAT observations for estimating $CO_2$ flux at regional or





finer scale. It is therefore important to investigate how current OCO-2 $XCO_2$ data differ from GO-
SAT $XCO_2$ data in constraining carbon budget.

In this study, we evaluate the performance of GOSAT and OCO-2 $XCO_2$ data in constraining

terrestrial ecosystem carbon flux. GOSAT and OCO-2 $XCO_2$ retrievals produced by the NASA At-
mospheric $CO_2$ Observations from Space (ACOS) team are applied to infer monthly terrestrial eco-
system carbon sinks and sources from Oct, 2014 to December, 2015, using a 4D-Var scheme based
on the GEOS-Chem Adjoint model. Inversion results are evaluated against surface flask $CO_2$ obser-
vations and TCCON $XCO_2$ measurements. We analyze the differences of inverted terrestrial ecosys-
tem carbon flux between using two $XCO_2$ data. The inverted carbon fluxes are also compared with
results from other datasets such as CarbonTracker CT2016 (Peters, et al., 2007) and Global Carbon
Project (GCP) 2017 (Le Quéré et al., 2018). This paper is organized as follows. Section 2 briefly
introduces GOSAT and OCO-2 $XCO_2$ retrievals and the inversion methodology and settings. Results
and discussions are presented in Section 3, and Conclusions are given in Section 4.

**2. Data and Method**
2.1 GOSAT and OCO-2 $XCO_2$ retrievals

Developed jointly by the National Institute for Environmental Studies (NIES), the Japanese

Space Agency (JAXA) and the Ministry of the Environment (MOE) of Japan, GOSAT was de-
signed to measure total column abundances of $CO_2$ and $CH_4$. The satellite flies at a 666 km altitude
in a sun-synchronous orbit with 98° inclination that crosses the equator at 12:49 local time. It co-
vers the whole globe in three days and has a footprint of 10.5 $km^2$ at nadir. OCO-2 is NASA's first
mission dedicated to measuring atmospheric $CO_2$ concentration. It flies at 705 km altitude in a sun-
synchronous orbit with an overpass time at approximately 13:30 local time and a repeat cycle of 16
days. Its grating spectrometer measures reflected sunlight in three near-infrared regions (0.765, 1.61
and 2.06 μm) to retrieve $XCO_2$. OCO-2 has a footprint of 1.29×2.25 $km^2$ at nadir and acquires eight



cross-track footprints creating a swath width of 10.3 km.
Both GOSAT and OCO-2 $XCO_2$ products were created using the same retrieval algorithm,
which is based on a Bayesian optimal estimation approach (Roggers et. al., 2000; O Dell et. al.,
2011). The GOSAT and OCO-2 $XCO_2$ data used in this study are Version 7.3 Level 2 Lite products
at the pixel level. The $XCO_2$ data from lite products are bias-corrected (Wunch et. al., 2011). Before
used in our inversion system, the data are processed in three steps. First, the retrievals for the glint
soundings over oceans have relatively larger uncertainty, thus the data over oceans are not used in
our inversions (Wunch et. al., 2017). Second, in order to achieve the most extensive spatial cover-
age with the assurance of using best quality data available, the $XCO_2$ data are filtered with two pa-
rameters, namely warn_levels and xco2_quality_flag, which are provided along with the $XCO_2$
data. All data with xco2_quality_flag not equaling 0 are removed, the rest are divided into three
groups according the value of warn_levels, namely group 1, group 2 and group 3. In group 1, the
warn_levels are less than 8, in group 2, the warn_levels are greater than 9 and less than 12, and in
group 3, those are greater than 13. Group 1 has the best data quality, followed by group 2, and
group 3 is the worst. Third, the pixel data are averaged within the grid cell of 2°×2.5°, which is the
resolution of the global atmospheric transport model used in this study. In each grid of 2°×2.5°,
only the groups of best data quality are selected and then averaged. The other variables like column
averaging kernel, retrieval error and so on which are provided along with the $XCO_2$ product are also
dealt with the same method.  Figures 1a and 1b show the coverages and data amount of GOSAT
and OCO-2 $XCO_2$ data during the study period after processing.





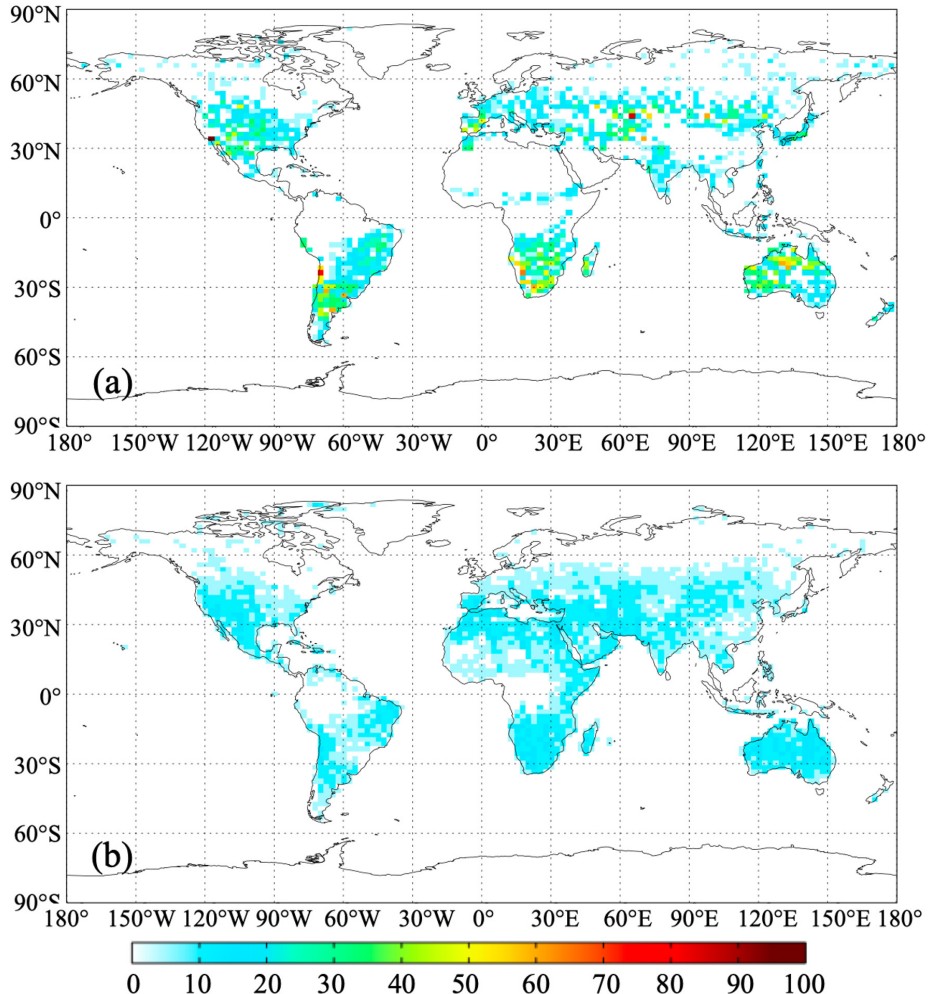


**Figure 1**. Data amount of each grid cell (2°×2.5°) of ACOS $XCO_2$ used in this study (a, GOSAT; b, OCO-2)

## 2.2 Surface flask and TCCON $CO_2$ measurements

We evaluate the inversion results through comparing the modeled $CO_2$ mixing ratios using the
posteriori fluxes with surface flask measurements and Total Carbon Column Observing Network
(TCCON) $XCO_2$ observations. The flask measurements of $CO_2$ mixing ratios are downloaded from
World Data Center for Greenhouse Gases (WDCGG) under the World Meteorological Organization
(WMO) Global Atmospheric Watch (GAW) programme (http://ds.data.jma.go.jp/gmd/wdcgg/). 47





surface sites which have valid observations for at least two months in 2015 are chosen. TCCON is a
network of ground-based Fourier Transform Spectrometers that measure direct near-infrared solar
absorption spectra. Column-averaged abundances of atmospheric constituents including $CO_2$, $CH_4$,
$N_2O$, HF, CO, $H_2O$, and HDO are retrieved through these spectra. We use $XCO_2$ retrievals from 13
stations from TCCON GGG2014 dataset (Blumenstock et al., 2017; Deutscher et al., 2017; Griffith
et al., 2017a, b; Kivi et al., 2017; Morino et al., 2017; Notholt et al., 2017a, b; Sherlock et al., 2017;
Sussmann and Rettinger, 2017; Warneke et al., 2017; Wennberg et al., 2017a, b). The locations of
47 flask sites and 13 TCCON stations are shown in Figure 2.

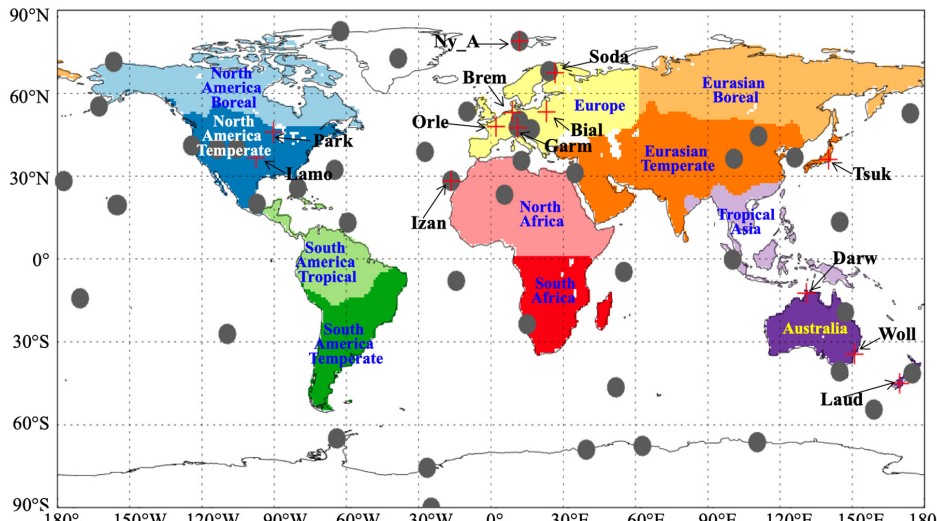

**Figure 2.** Distributions of the 47 surface flask sites (gray solid circle) and 13 TCCON sites (red
cross mark), shaded shows the 11 TRANSCOM regions
**2.3 GEOS-Chem 4DVAR assimilation framework**
In this work, bias-corrected $XCO_2$ retrievals are assimilated to estimate monthly terrestrial eco-
system carbon fluxes using the GEOS-Chem and its adjoint model in a 4D-Var assimilation frame-
work.
**2.3.1 GEOS-Chem model**
GEOS-Chem model (http://geos-chem.org) is a global three-dimensional chemistry transport



model (CTM), which is driven by assimilated meteorological data from the Goddard Earth Observ-
ing System (GEOS) of the NASA Global Modeling and Assimilation Office (GMAO) (Rienecker et
al., 2008). The original $CO_2$ simulation in the GEOS-Chem model was developed by Suntharalin-
gam *et al.* (2004) and accounts for $CO_2$ fluxes from fossil fuel combustion and cement production,
biomass burning, terrestrial ecosystem exchange, ocean exchange and biofuel burning. Nassar et al.
(2010) updated the $CO_2$ simulation with improved inventories. In addition to the inventories in ear-
lier version, the new $CO_2$ fluxes includes $CO_2$ emissions from international shipping, aviation (3D)
and the chemical production of $CO_2$ from CO oxidation throughout the troposphere.  In most other
models, the oxidation of CO was treated as direct surface $CO_2$ emissions. The details of the $CO_2$
simulation and the $CO_2$ sinks/sources inventories could be found in Nassar et al. (2010). The ver-
sion of GEOS-Chem model used in this study is v8-02-01.
**2.3.2 GEOS-Chem adjoint model**
An adjoint model is used to calculate the gradient of a response function of one model scalar
(or cost function) with respect to a set of model parameters. The adjoint of the GEOS-Chem model
was first developed for inverse modeling of aerosol (or their precursors) and gas emissions (Henze
et al., 2007). It has been implemented to constrain sources of species such as CO, $CH_4$, and $O_3$ with
satellite observations (Kopacz et al., 2009, 2010; Jiang et al., 2011; Wecht et al., 2012; Parrington et
al., 2012). Several studies have successfully used this adjoint model to constraint carbon sources
and sinks with surface flask measurements of $CO_2$ mixing ratio and space-based $XCO_2$ retrievals
(Deng et al., 2014; Liu et al., 2014; Deng et al., 2016; Liu et al., 2017).
**2.3.3 Inversion method**
In the GEOS-Chem inverse modeling framework, the 4D-Var data assimilation technique is
employed for combining observations and simulations to seek a best optimal estimation of the state
of a system. This approach seeks the scaling factors of the carbon flux that minimize the cost func-
tion, J, given by:



$$J(c) = \frac{1}{2}\sum_{i=1}^{N}\left(XCO_{2,i}^{m} - XCO_{2,i}^{obs}\right)S_{obs,i}^{-1}\left(XCO_{2,i}^{m} - XCO_{2,i}^{obs}\right) + \frac{1}{2}(c - c_a)S_c^{-1}(c - c_a)$$

where N is total number of satellite $XCO_2$ observations; $XCO_2^m$ and $XCO_2^{obs}$ are modeled and ob-
served total column averaged dry air mole faction of $CO_2$ respectively; $c_a$ is the priori scaling factor
of the carbon flux, which is typically set as unity; $S_{obs}$ is the model-data mismatch error covariance
matrix; $S_c$ is the scaling factor error covariance matrix. The gradients of the cost function with re-
spect to scaling factors calculated with the adjoint model are supplied to an optimization routine
(the L-BFGS-B optimization routine; Byrd et al., 1995; Zhu et al., 1994), and the minimum of the
cost function is sought iteratively.
For the modeled $CO_2$ column to be comparable with the satellite $XCO_2$ retrievals, the modeled
$CO_2$ concentration profile should be first mapped into the satellite retrieval levels and then convo-
luted with retrieval averaging kernels. The modeled $XCO_2$ is computed by:
$$XCO_2^m = XCO_2^a + \sum_j h_j a_j(A(x) - y_{a,j})$$

where $j$ denotes retrieval level, $x$ is the modeled $CO_2$ profile; $A(x)$ is a mapping matrix; $XCO_2^a$ is a
priori $XCO_2$, $h_j$ is pressure weighting function, $a_j$ is the satellite column averaging kernel and $y_a$ is
the priori $CO_2$ profile for retrieval.
**3. Inversion settings**
In this study, the GEOS-Chem model was run in a horizontal resolution of 2°×2.5° for 47 verti-
cal layers. Two experiments, one with GOSAT data and the other with OCO-2 data, were conducted
from Oct 1, 2014 to December 31, 2015. The posteriori dry air mole fraction of $CO_2$ of Oct 1, 2014
from CT2016 product was taken as the initial concentration. The first three months were taken as
the spin-up period. The priori carbon fluxes used in this study include: fossil fuel and cement manu-
facture $CO_2$ emissions from the Carbon Dioxide Information Analysis Center (CDIAC) (Andres et





al., 2011), biomass burning $CO_2$ emissions from the Global Fire Emissions Database version 4.1
(GFEDv4) (van der Werf et al., 2010; Giglio et al., 2013) ; terrestrial ecosystem carbon exchanges
from the Carnegie-Ames-Stanford Approach (CASA) model GFED4.1 simulation (Potter el al.,
1993; van der Werf et al., 2010 ) ; $CO_2$ exchanges over the ocean surface from the posteriori air-sea
$CO_2$ flux from CT2016 (Peters *et al.* 2007, with updates documented at http://carbon-
tracker.noaa.gov); monthly shipping emissions of $CO_2$ from the International Comprehensive
Ocean–Atmosphere Data Set (ICOADS) (Corbett and Koehler, 2003; Endresen et al., 2004, 2007);
3-D aviation $CO_2$ emissions (Kim et al., 2007; Wilkerson et al., 2010); and 3-D chemical produc-
tion of $CO_2$ from the oxidation of other atmospheric carbon species and its surface corrections (Nas-
sar et al., 2010). It is noted that the terrestrial ecosystem $CO_2$ exchanges, fossil fuel and cement
manufacture emissions and biomass burning emissions in our inversions are the same as those in
CT2016. Since we exclude $XCO_2$ retrievals over ocean, in our inversions, the terrestrial ecosystem
exchanges might compensate for the under-constrained ocean $CO_2$ fluxes. To mitigate the impact of
the lack of $XCO_2$ observations over ocean, we directly use the posteriori ocean $CO_2$ fluxes of
CT2016, which were well constrained with surface flask observations at ocean sites.  Only terres-
trial ecosystem $CO_2$ exchanges are optimized in our inversions.

An efficient computational procedure for constructing non-diagonal priori flux error covariance

matrix which accounts for the spatial correlation of errors is implemented (Single et al, 2011). The
construction is based on the assumption of exponential decay of error correlations. Other than form-
ing covariance matrix explicitly, multiple-dimensional correlations are represented by tensor prod-
ucts of one dimensional correlation matrices along longitude and latitudinal directions. For the two
inversions, the scale lengths assigned along longitudinal and latitudinal directions are 500 km and
400 km for terrestrial ecosystem exchange and 1000 km and 800 km for ocean exchange, respec-
tively. No correlations between different types of fluxes are assumed. The temporal correlations are
also neglected. Global annual uncertainty of 100% and 40% are assigned for terrestrial ecosystem



and ocean CO₂ exchanges, respectively (Deng and Chen, 2011). Accordingly, the ratios of uncer-
tainty to the priori land and ocean fluxes in each month at the grid cell level are assigned with 5 and
3, respectively.
The observation error covariance matrix is constructed using the retrieval error, which is pro-
vided along with the XCO₂ data. Observation errors are assumed to be uncorrelated at model grid
level. To account for the correlated observation errors, as shown in section 2.1, the pixel level re-
trieval errors are filtered and averaged to model at the grid level, and then inflated by a factor of 1.9
to ensure the chi-square testing of $\chi^2$ value to be close to 1 (Tarantola, 2004; Chevallier et al.,

2007) .

**4. Results and Discussions**
4.1 **Global carbon budget**
Table 1 presents the optimized global carbon budgets by the two experiments in 2015. For com-
parison purposes, the prior fluxes used in this study and the estimates of CT2016 and GCP2017 are
also shown in Table 1. The optimized global land sinks based on GOSAT and OCO-2 XCO₂ retriev-
als are -3.48 and -2.94 PgC yr⁻¹, respectively, which are larger than the prior value, but lower than
the CT2016 estimate based on the flask/in-situ CO₂ concentration observations. The differences of
ocean fluxes among a priori and two inversions are small since we don't assimilate XCO₂ data over
ocean. GCP gives a comprehensive estimate for the global carbon budget every year. In the GCP
2017 estimates, the components of the global carbon budget include fossil fuel and industry, land-
use change emissions, atmospheric growth, ocean sink, land sink, and budget imbalance, which are
different from those in this study and CT2016 (Table 1). For ease of comparison, the budget imbal-
ance, land sink and land-use change emissions are combined as land net flux, and then the biomass
burning emissions and the land sink in this study and those from CT2016 are combined to obtain
the land net flux. As shown in Table 1, the prior estimate gives the smallest net land flux (-0.5 Pg C
yr⁻¹), and the CT2016 estimate is the largest (-1.7 PgC yr⁻¹). The land net flux optimized based on





GOSAT and OCO-2 $XCO_2$ retrievals fall in between (-0.74 and -1.28 PgC yr$^{-1}$, respectively), and
are much closer to the estimate of GCP 2017 (-1.03 PgC yr$^{-1}$). A global net flux from GCP is in-
ferred from the global annual atmospheric carbon growth rate, which represents relatively accu-
rately the net carbon flux added into atmosphere. It could be found that the global net flux esti-
mated based on GOSAT data is the closest to the GCP estimate, while the one estimated using
OCO-2 data is higher and the CT2016 estimate is much lower than the GCP result, indicating that
the land and ocean carbon uptakes in CT2016 were overestimated, while those optimized using
OCO-2 data might be underestimated.
**Table 1**. Global carbon budgets estimated by the OCO-2 experiment, GOSAT experiment in this
study as well as those from the priori fluxes, CT2016 and GCP2017 (PgC yr$^{-1}$)

|  | Priori | OCO-2 experiment | GOSAT experiment | CT2016 | GCP2017 |
|---|---|---|---|---|---|
| Fossil fuel and industry | 9.83 | 9.83 | 9.83 | 9.83 | 9.83 |
| Biomass burning emissions | 2.2 | 2.2 | 2.2 | 2.2 | 1.52[a] |
| Land sink | -2.5 | -2.94 | -3.48 | -3.9 | -2.55[b] |
| Land net flux | -0.5 | -0.74 | -1.28 | -1.7 | -1.03 |
| Ocean sink | -2.41 | -2.43 | -2.45 | -2.41 | -2.57 |
| Global net flux | 7.12 | 6.66 | 6.1 | 5.72 | 6.23[c] |

[a] land-use change emissions in GCP2017
[b] land sink plus budget imbalance
[c] atmospheric growth
## 4.2 Regional carbon flux
Figure 3 shows the distributions of annual land and ocean carbon fluxes (excluding fossil fuel
and biomass burning carbon emissions, same thereafter) of the prior and the estimates using GOSAT
and OCO-2 data. It could be found that compared with the prior fluxes, the carbon sinks in Central
America, south and northeast China, east and central Europe, south Russia and east Brazil are obvi-
ously increased in GOSAT inversion. Except for east Brazil, the land sinks in those areas in OCO-2
inversion are also increased, but much weaker than those in GOSAT inversion, and in east Brazil, it



turns to a significant carbon source. In contrast, in east and central Canada, north Russia, north Eu-
rope, west Indo-China Peninsula, north Democratic Republic of the Congo and west Brazil, their
carbon sources are significantly increased in both GOSAT and OCO-2 inversions. In east and central
Canada, north Europe and west Brazil, there are much stronger carbon sources in OCO-2 inversion.

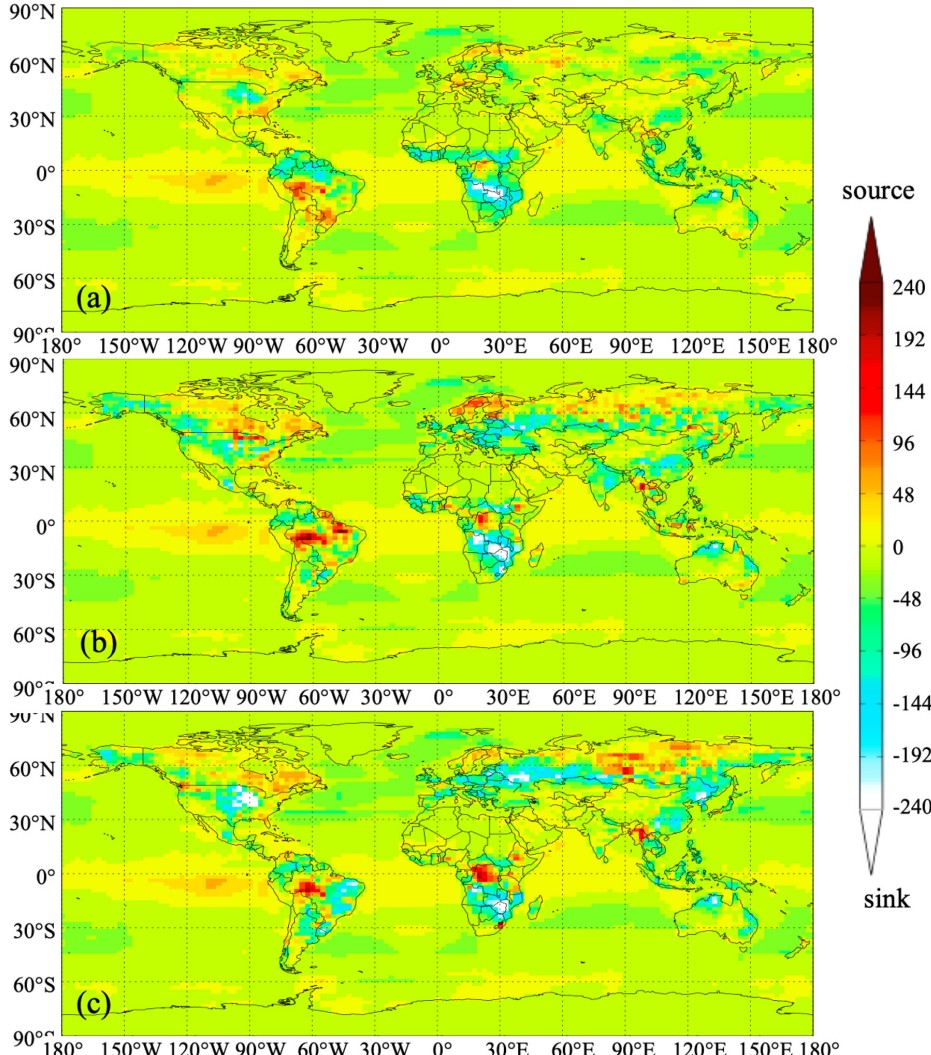

**Figure 3**. Distributions of annual land and ocean carbon fluxes a) priori flux and posteriori fluxes
based on (b) OCO-2 and (c) GOSAT data (gC m$^{-2}$yr$^{-1}$)

To better investigate the differences between GOSAT and OCO-2 inversions as well as their



differences with the prior fluxes, we aggregate the prior and inferred land fluxes into 11 TRANSCOM
land regions (ref.??). Figure 4 shows aggregated annual land surface fluxes from the prior, GOSAT
and OCO-2 inversions for the 11 land regions. For comparison purposes, the optimized surface fluxes
in CT2016 are also aggregated and shown in Figure 4.

Clearly, in most regions, the land sinks inverted based on GOSAT data are stronger than those

inferred from OCO-2 data. In Northern Temperate and Southern Temperate Lands (i.e., North Amer-
ica Temperate, Eurasian Temperate, Europe, South America Temperate, South Africa and Australia),
except for Eurasian Temperate and South Africa, the inferred land sinks using GOSAT $XCO_2$ are
much stronger than those optimized using OCO-2 data. For example, in North America Temperate,
the carbon sink of GOSAT experiment (-0.5 Pg C yr$^{-1}$) is about twice that of the OCO-2 inversions (-
0.27 Pg C yr$^{-1}$); and in South America Temperate, the estimated land sink based on GOSAT data (-
0.47 Pg C yr$^{-1}$) is about 4 times as large as the OCO-2 inversions (-0.12 Pg C yr$^{-1}$). For the total
Temperate Land, the optimized land sinks based on GOSAT and OCO-2 $XCO_2$ retrievals are -2.95
and -2.59 Pg C yr$^{-1}$, respectively (Table 2). In the tropics (i.e., South America Tropical, North Africa,
and Tropical Asia), except for North Africa, the carbon sinks inferred from GOSAT data are also
larger than those estimated using OCO-2 data. In Tropical Asia, the estimated land sink based on
GOSAT data (-0.28 Pg C yr$^{-1}$) is about 2 times of the OCO-2 inversions (-0.13 Pg C yr$^{-1}$); in South
America Tropical, the OCO-2 inversion result is a carbon source of 0.19 PgC yr$^{-1}$, while GOSAT
inversion gives a weak sink of -0.05 Pg C yr$^{-1}$. The total carbon sinks in tropical land inverted from
GOSAT and OCO-2 data are -0.36 and -0.20 Pg C yr$^{-1}$, respectively. In Northern Boreal Land, in-
cluding North America Boreal and Eurasian Boreal, the total carbon sinks inverted with GOSAT (-
0.18 Pg C yr$^{-1}$) and OCO-2 (-0.16 Pg C yr$^{-1}$) data are comparable. However, the two $XCO_2$ data have
opposite performances in these two areas, namely in Eurasian Boreal, the inverted land sink with
GOSAT is stronger than that with OCO-2; while in North America Boreal, it is the opposite.





For different continents (Table 2), in Asia and Australia, their carbon sinks inverted from GO-

SAT and OCO-2 data are comparable. In North America, South America and Europe, the land sinks
in GOSAT inversion are much stronger than those in OCO-2 inversion. Especially in South America,
the GOSAT inversion result is a strong carbon sink (-0.51 Pg C yr$^{-1}$), while in OCO-2 inversion, it is
a weak carbon source (0.06 Pg C yr$^{-1}$). Conversely, in Africa, the land sink estimated with GOSAT
data is much weaker than those from OCO-2 data, the former (-0.59 Pg C yr$^{-1}$) being only about the
half of the latter (-1.13 Pg C yr$^{-1}$).

Compared with the prior fluxes, basically, the inferred land fluxes in Northern Temperate re-

gions have largest changes, followed by those in Tropical regions and Southern Temperate lands,
while in boreal regions, the changes are weakest. Basically, in temperate regions, the inferred land
sinks are significantly increased, while those in tropical regions are decreased. In boreal regions, the
changes using two XCO$_2$ data are not consistent. As shown in Table 3, in Northern Temperate Land,
the increases of carbon sinks constrained by OCO-2 and GOSAT reach to 1.03 and 1.33 Pg C yr$^{-1}$,
respectively, while in Tropical Land, the enhancements of carbon sources are 0.82 and 0.66 Pg C yr$^{-}$
$^{1}$, respectively, whereas in Northern Boreal Land, the changes caused by the two XCO$_2$ data are only
0.005 and -0.015 Pg C yr$^{-1}$. For different TRANSCOM regions and different XCO$_2$ used, the changes
of carbon fluxes have large differences. When using GOSAT data, Europe has the largest change in
the carbon flux (-0.63 Pg C yr$^{-1}$), followed by South America Temperate (-0.50 Pg C yr$^{-1}$) and North
America Temperate (-0.41 Pg C yr$^{-1}$); when using OCO-2 data, the largest carbon sink changes are in
South America Tropical (0.46 Pg C yr$^{-1}$) and Eurasian Temperate (-0.46 Pg C yr$^{-1}$), followed by Eu-
rope (-0.39 Pg C yr$^{-1}$). Since the same setup used in these two inversions and the same algorithm
adopted for retrieving XCO$_2$ from GOSAT and OCO-2 measurements, the different impacts of XCO$_2$
data on land sinks may be related to the spatial coverage and the amount of data in these two XCO$_2$
datasets. As shown in Figure 1, in different latitude zones, the spatial coverage and the data amount
of GOSAT and OCO-2 have large differences. Statistics show that basically, the amount of data is





largest in northern temperate land, followed by southern temperate land and tropical land, and least
in northern boreal regions, corresponding to the magnitude of changes of carbon fluxes in these zones.
For one specific zone, the different impacts of these two $XCO_2$ datasets may be also related to their
data amount. For example, in northern temperate land, GOSAT has more $XCO_2$ data than OCO-2.
Accordingly, the change of carbon flux caused by GOSAT is larger than that caused by OCO-2. Con-
versely, in Tropical Land, OCO-2 has more data than GOSAT, and as shown before it has more sig-
nificant impact on the land sink. This relationship could also be found in each TRANSCOM region.
Figure 5 gives a relationship between the $XCO_2$ data amount ratios of GOSAT to OCO-2 and the land
sinks absolute change ratios caused by GOSAT to OCO-2 for 11 TRANSCOM land regions. Obvi-
ously, except for North and South Africa, there is a significant linear correlation (R=0.95) between
these two ratios, suggesting that the more $XCO_2$ data, the more carbon flux is changed. In North
Africa, we find that OCO-2 has better spatial coverage and more data than GOSAT, as shown in
Figure 1, although the differences mainly occur in the Sahara where the carbon flux is very weak, but
near the equatorial region where the carbon flux is large, OCO-2 still has more data than GOSAT; in
southern Africa, both $XCO_2$ have good spatial coverage, the amount of GOSAT data is about 1.5
times that of OCO-2, but the changes in the carbon flux caused by GOSAT is about 10 times that of
OCO-2. This indicates that in addition to the spatial coverage and the amount of data, the instrument
characteristics such as sensor accuracy and spatial resolution may also have impact on the inversion
results.

Compared with the CT2016 results, in temperate regions, except for Australia and Europe, the

carbon sinks estimated from the two $XCO_2$ datasets in this study are basically comparable with those
inferred based on surface observations in CT2016. In Australia and Europe, the inverted carbon sinks
with $XCO_2$ data in this study are both much stronger than CT2016. Especially in Europe, the CT2016
estimate is a significant source of 0.26 Pg C, whereas our inversions suggest a strong carbon sink
ranging between -0.40 and -0.63 Pg C $yr^{-1}$. Although previous studies (Basu et al., 2013; Chevallier



et al., 2014; Deng et al., 2014; Houweling et al., 2015) also showed enhanced carbon uptake in Europe
using GOSAT data, but until now, the strong Europe uptake as inferred from satellite data is still in
intense debate (Reuter et al., 2014; Feng et al., 2016; Reuter et al., 2017). Examination of the seasonal
variation of the inverted fluxes in Europe shows that during the growing season, both CT2016 and
our inversions estimate similar carbon uptake, whereas in the non-growing season, CT2016 produces
more carbon release than our inversions. Since there are few satellite measurements in Europe during
the dormant season, the carbon release by respiration might not be well constrained by satellite data
but close to the prior value.

In boreal regions, the inverted land sinks of CT2016 are significantly stronger than those in this

study, especially in the Eurasian Boreal, where the carbon uptake estimated by CT2016 is almost 1
Pg C yr$^{-1}$ larger than our estimates. One explanation for the large sink in CT2016 in this area is that
there is a mutual compensation for carbon sinks between Europe and Eurasian Boreal because of the
large differences in the amount of observations between these two areas. Kim et al. (2017) found that
with the addition of Siberian in-situ measurements to their inversion system, the carbon uptake in
Europe was enhanced while it decreased in the boreal Eurasian. Saeki et al. (2013) also reported that
more $CO_2$ observations over Siberian used in the inversion system would weaken the summer carbon
uptake in this area. These studies indicate that for CT2016, the carbon sink in Europe may be under-
estimated, while in boreal Eurasian, it may be significantly overestimated. Since there are only very
few $XCO_2$ observations from both GOSAT and OCO-2 in this area, our estimates are very close to
the prior and much weaker than those found by Saeki et al. (2013) and Kim et al. (2017), indicating
that the land sink in the Eurasian Boreal is underestimated in this study. Combined the land sinks of
Europe and Eurasian Boreal, the inverted land sinks of GOSAT (-0.86 Pg C yr$^{-1}$) is comparable with
CT2016 (-0.92 Pg C yr$^{-1}$), indicating that the land sink in Europe inferred from GOSAT may be also
overestimated to a certain extent.

In the tropical regions, the inverted land sinks of CT2016 are also all much stronger than our



estimates, but the differences between the inverted and the prior fluxes in CT2016 are significantly
smaller than those in this study, mainly because of the lack of surface $CO_2$ observations in tropical
areas, especially in Tropical Asia.
For different continents (Table 2), the largest difference between this study and CT2016 is found
in Asia. Using an ensemble of seven atmospheric inverse systems, Thompson et al. (2016) reported
that the Asian land biosphere $CO_2$ flux (including land-use change and fires) was a net sink of −0.46
(−0.70~0.24) Pg C yr$^{-1}$ (median and range) for 1996–2012. The land biosphere $CO_2$ fluxes (also in-
cluding biomass burning emissions) estimated based on OCO-2 and GOSAT in this study are -0.37
and -0.42 Pg C yr$^{-1}$, respectively, which are comparable with the result of Thompson et al. (2016).

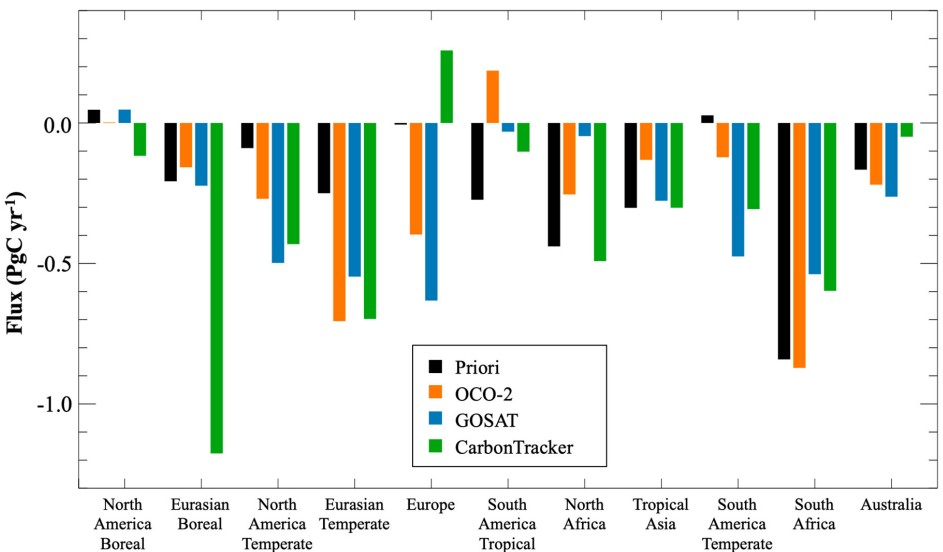


**Figure 4**. Aggregated annual land fluxes of the 11 TRANSCOM land regions






**Table 2**. The priori and posteriori fluxes in six continents and boreal, temperate and tropical lands

| Regions | Prior | OCO-2 | GOSAT | CT2016 |
|---|---|---|---|---|
| North America | -0.04 | -0.27 | -0.45 | -0.55 |
| South America | -0.25 | 0.06 | -0.51 | -0.41 |
| Europe | -0.01 | -0.40 | -0.63 | 0.26 |
| Asia | -0.76 | -0.99 | -1.05 | -2.18 |
| Africa | -1.28 | -1.13 | -0.59 | -1.09 |
| Australia | -0.17 | -0.22 | -0.26 | -0.05 |
| Northern Boreal Land | -0.16 | -0.16 | -0.18 | -1.29 |
| Northern Temperate Land | -0.35 | -1.37 | -1.68 | -0.87 |
| Tropical Land | -1.01 | -0.20 | -0.36 | -0.90 |
| Southern Temperate Land | -0.98 | -1.21 | -1.28 | -0.95 |


**Table 3**. Differences between the inferred and the prior carbon fluxes, and the data amount of $XCO_2$
in different regions

| Region | Differences (Pg C yr$^{-1}$) | | $XCO_2$ data amount | |
|---|---|---|---|---|
| | OCO-2 | GOSAT | OCO-2 | GOSAT |
| North America Boreal | -0.05 | 0.00 | 1143 | 639 |
| North America Temperate | -0.18 | -0.41 | 2390 | 3163 |
| South America Tropical | 0.46 | 0.24 | 800 | 421 |
| South America Temperate | -0.15 | -0.50 | 1711 | 3500 |
| North Africa | 0.19 | 0.39 | 3208 | 674 |
| South Africa | -0.03 | 0.30 | 2057 | 3060 |
| Eurasian Boreal | 0.05 | -0.02 | 1714 | 1339 |
| Eurasian Temperate | -0.46 | -0.30 | 5323 | 4782 |
| Tropical Asia | 0.17 | 0.03 | 726 | 550 |
| Australia | -0.05 | -0.10 | 2011 | 3110 |
| Europe | -0.39 | -0.63 | 1604 | 2106 |
| Global land | -0.44 | -0.98 | 22687 | 23344 |
| Northern Boreal Land | 0.005 | -0.02 | 2857 | 1978 |
| Northern Temperate Land | -1.03 | -1.33 | 9317 | 10051 |
| Tropical Land | 0.82 | 0.66 | 4734 | 1645 |
| Southern Temperate Land | -0.23 | -0.30 | 5779 | 9670 |




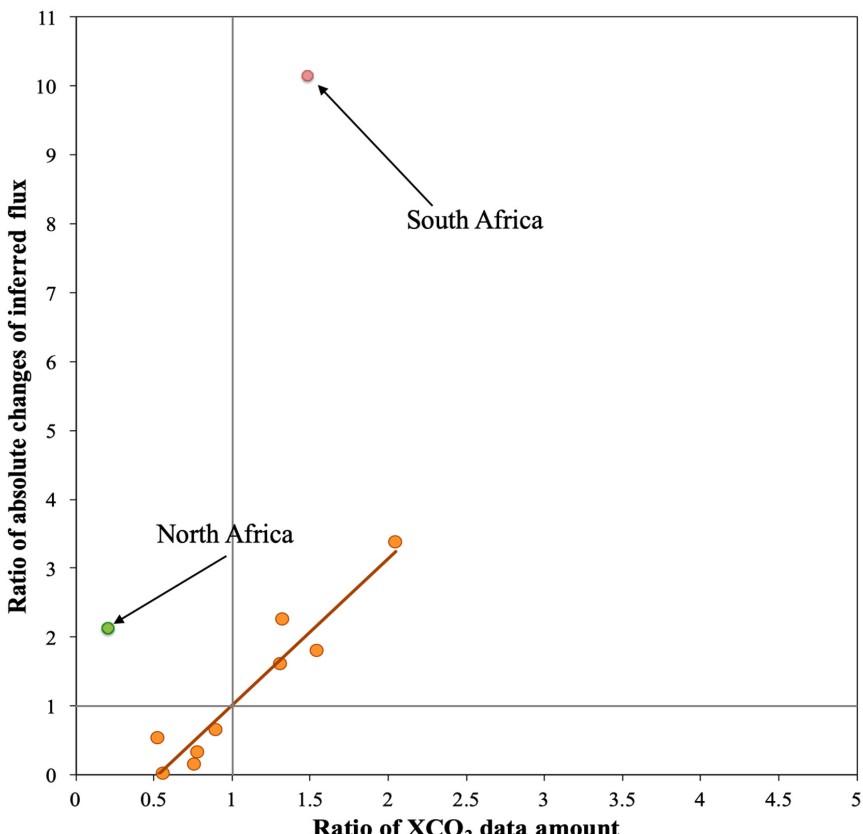


**Figure 5.** Scatter plot for the ratio of GOSAT to OCO-2 $XCO_2$ data amount versus the ratio of abso-
lute changes of the land sinks caused by GOSAT to OCO-2 in the 11 TRANSCOM land regions

**4.3 Evaluation for the inversion results**

**4.3.1 Flask observations**

We use flask observations from 47 surface sites (Figure 2) to evaluate the posterior fluxes. The

GEOS-Chem model is driven with the prior flux and the two posterior fluxes to obtain the prior and

posterior $CO_2$ mixing ratios. The simulated $CO_2$ mixing ratios are sampled at each observation site

and within half an hour of observation time. Figure 6 shows a summary of comparisons of the simu-

lated $CO_2$ mixing ratio against the flask measurements. The mean difference between the prior $CO_2$

mixing ratio and the flask measurements is 1.14 ppm, with a standard deviation of 2.73 ppm. Using





posterior fluxes inferred from GOSAT and OCO-2 data, the mean differences between the posterior
$CO_2$ mixing ratio and flask measurements are reduced to -0.32 and 0.48 ppm, with a standard devia-
tion of 2.59 and 2.79 ppm, respectively. It is noted that the mean difference in the posterior $CO_2$
mixing ratio between GOSAT and OCO-2 inversions at 47 observation sites is 0.8 ppm, much larger
than the 0.26 ppm (0.56 Pg C/2.12 Pg C/ppm=0.26 ppm) difference between their global net flux
estimates, indicating the relatively large difference in posterior flux between these two inversions at
the regional scale. Figure 7 shows the biases at each observation site in different latitudes. It could be
found that the biases between the simulations and the observations in the northern hemisphere are
significantly larger than those in southern hemisphere since the carbon flux distribution of the north-
ern hemisphere is more complex than that of the southern hemisphere. When the prior flux is used,
almost all sites in the northern hemisphere have significant positive deviations, with an average of
1.7 ppm, while in the southern hemisphere, the deviations are very small, with an average bias of
only 0.08 ppm; when using the posteriori flux of OCO-2, the deviations in most northern hemisphere
sites are significantly reduced, with the average deviation falling to half the original, to 0.85 ppm,
while in the southern hemisphere, at most sites, the biases increase by variable amounts, with a mean
of -0.13 ppm; when using the posterior flux of GOSAT, the deviations are further reduced to -0.04
ppm in the northern hemisphere but further increased to -0.55 ppm in the southern hemisphere. These
suggest that GOSAT and OCO-2 data can effectively improve the carbon fluxes estimate in the north-
ern hemisphere, but overestimate the land sinks in the southern hemisphere, especially for GOSAT.





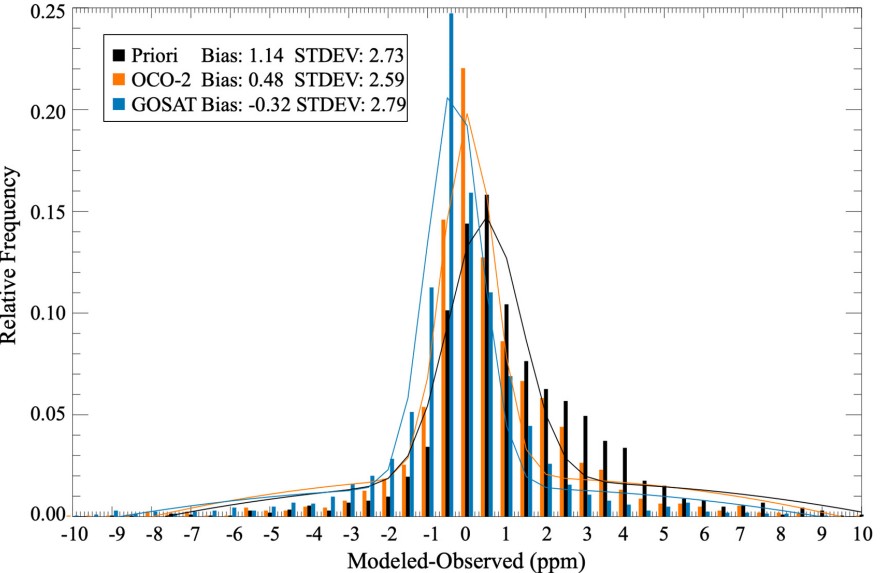

**Figure 6**. Statistical distribution of the modeled and observed mismatch errors in 47 surface flask
sites

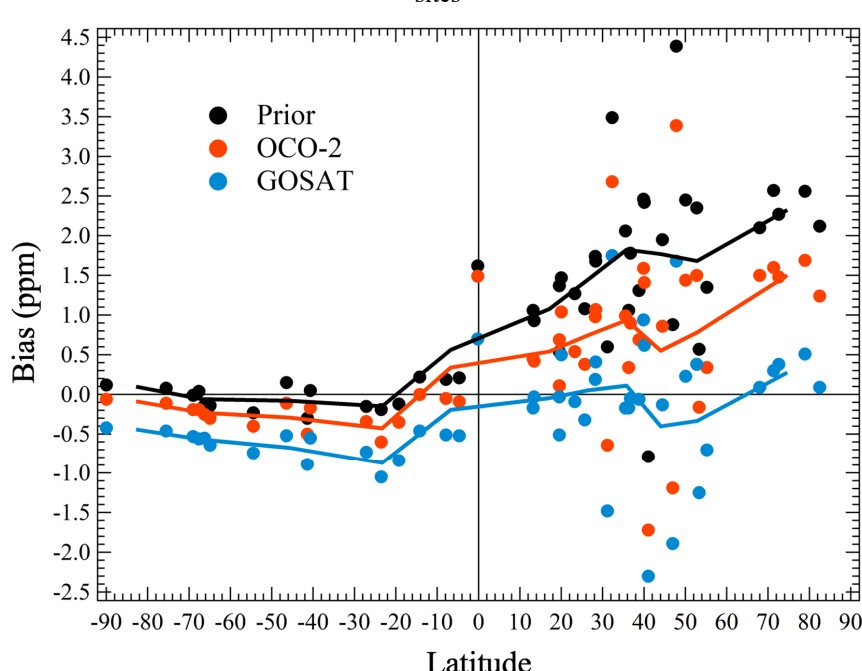

**Figure 7**. Biases of the simulated $CO_2$ mixing ratios against the flask measurements in different lat-
itudes (positive/negative biases represent modeled concentration being greater/less than the ob-
served)





### 4.3.2 TCCON observations


We also use ground $XCO_2$ observations from 13 TCCON sites (Figure 2) to evaluate our inver-
sion results. The simulated $CO_2$ concentrations at 47 vertical levels are mapped into 71 TCCON
levels. Following the approach of Wunch et al. (2011), using prior profiles and the averaging kernel
from the TCCON dataset, we calculated the modeled $XCO_2$ values at 13 TCCON sites. Figure 6
shows the comparison of modeled $XCO_2$ with TCCON observations. The mean difference between
prior $XCO_2$ and TCCON observations is 1.37 ppm, with a standard deviation of 1.25 ppm. Through
OCO-2 inversion, the mean difference between modeled and observed $XCO_2$ is slightly reduced to
0.93 ppm, with a standard deviation of 1.07 ppm, and through GOSAT inversion, the mean differ-
ence between modeled and observed $XCO_2$ is significantly reduced to 0.04 ppm with a standard de-
viation of 1.09 ppm. Figure 9 shows the bias at each TCCON site. Obviously, the biases at all
TCCON sites are positive when using the prior fluxes, ranging between 0.3 and 2.6 ppm.  The bi-
ases at the sites in the northern temperate and boreal areas are all above 1.5 ppm except for the La-
mont site. After using the posterior fluxes of OCO-2, the biases of all sites are reduced by 30%. Af-
ter using the posteriori fluxes of GOSAT, the biases are significantly reduced, ranging between -
0.48 and 1.03 ppm. For two of the three TCCON sites in the southern hemisphere, the biases are
changed to negative values when using the posteriori fluxes from GOSAT data, further indicating
the overestimation of carbon sinks by GOSAT data in the southern hemisphere.



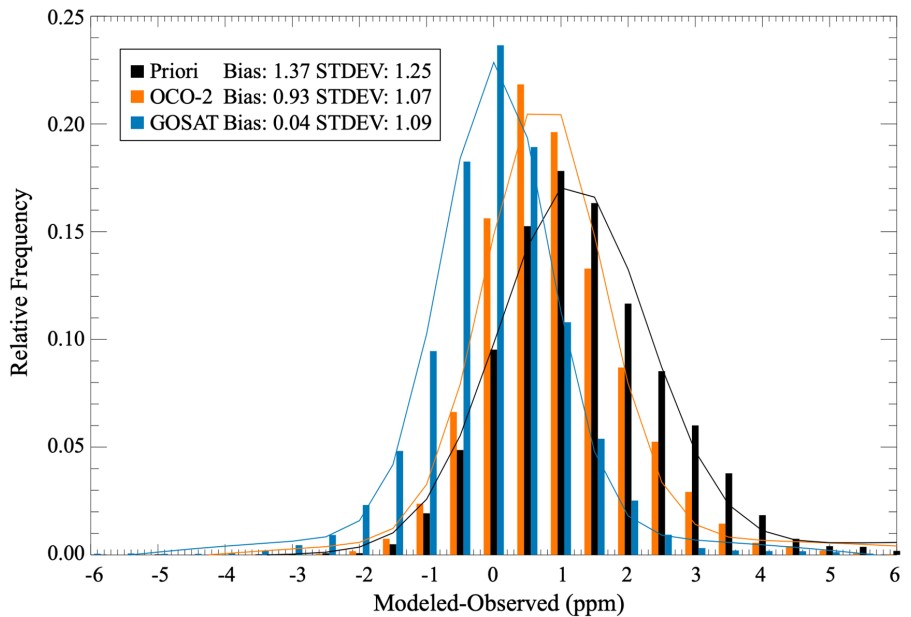


**Figure 8**. Statistical distributions of modeled and observed mismatch errors at 13 TCCON sites

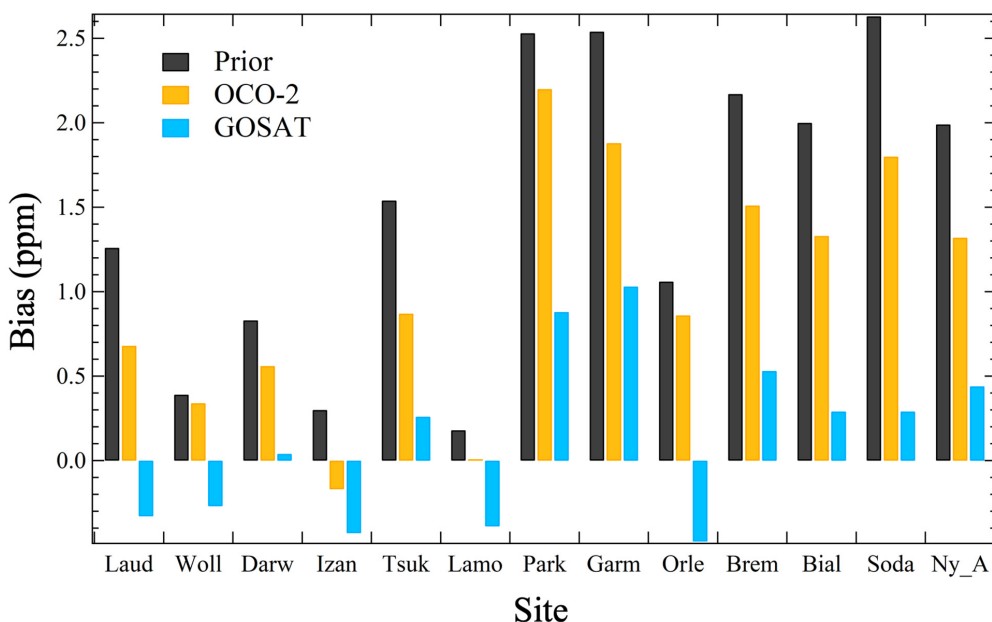


**Figure 9**. The biases between the modeled and observed $XCO_2$ at the 13 TCCON sites





## 5. Summary and Conclusions


In this study, we use both GOSAT and OCO-2 $XCO_2$ retrievals to constrain terrestrial ecosys-
tem carbon fluxes from Oct 1, 2014 to Dec 31, 2015, using a 4D-Var system within the GEOS-Chem
adjoint model. The posterior carbon fluxes estimated from GOSAT and OCO-2 data at both global
and regional scales during Jan 1 to Dec 31, 2015 are shown and discussed. Surface $CO_2$ mixing ratios
from 47 surface flask sites and $XCO_2$ observations from 13 TCCON sites are used to evaluate the
inversions of carbon fluxes using GOSAT and OCO-2 data.
Globally, the land net flux (including fossil fuel and biomass burning emissions) inferred from
GOSAT and OCO-2 $XCO_2$ retrievals are larger than the prior value, and lower than the estimate of
CT2016, but much closer to the estimate of GCP 2017. The terrestrial ecosystem carbon sink (ex-
cluding biomass burning emissions) estimated from GOSAT data is stronger than that inferred from
OCO-2 data, and the annual atmospheric $CO_2$ growth rate (global net flux) estimated based on GO-
SAT data is more consistent with the GCP estimate than that based on OCO-2. Regionally, in most
regions, the land sinks inverted based on GOSAT data are also stronger than those inferred from
OCO-2 data. Compared with the prior fluxes, basically, the inferred land sinks are significantly in-
creased in northern and southern temperate regions, and decreased in tropical regions. In addition,
the inferred carbon fluxes have the largest changes in Northern Temperate regions, followed by Trop-
ical and Southern Temperate regions, and the weakest in boreal regions. The different impact of $XCO_2$
on the carbon fluxes in different regions are mainly related to the spatial coverage and the amount of
$XCO_2$ data. Generally, a larger amount of $XCO_2$ data in a region is corresponding to a larger change
in the inverted carbon flux in the same region. Compared with the CT2016 results, the carbon sinks
optimized using $XCO_2$ in this study are comparable with CT2016 in most temperate regions, but
much weaker than CT2016 in the boreal and tropical regions.
Evaluations of the inversions using $CO_2$ concentrations from flask and TCCON measurements
showed that both posterior carbon fluxes of OCO-2 and GOSAT could significantly improve the





modeling of atmospheric $CO_2$ concentrations, and both the simulated surface $CO_2$ mixing ratio and
$XCO_2$ concentrations with GOSAT posterior fluxes are much closer to the observations than those
with OCO-2. Generally, in the northern hemisphere, the deviations are significantly reduced, while
in the southern hemisphere, the biases are elevated to a certain extent. These suggest that GOSAT and
OCO-2 data can effectively improve the carbon fluxes estimate in the northern hemisphere, while in
the southern hemisphere and some northern temperate regions, the optimized carbon sinks may be
overestimated.
**Author contributions**
FJ and HW designed the research, HW conducted inverse modeling, HW and FJ conducted data anal-
ysis and wrote the paper, JW, WJ and JC participated in the discussion of the results and provided
input on the paper for revision before submission.
**Competing interests**
The authors declare that they have no conflict of interest.
**Acknowledgements**
This work is supported by the National Key R&D Program of China (Grant No: 2016YFA0600204), National
Natural Science Foundation of China (Grant No: 41571452), and the Fundamental Research Funds for the
Central Universities (Grant No: 090414380021).

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
