# Peer review of "Differences of the inverted terrestrial ecosystem carbon flux between using GO- SAT and OCO-2 XCO₂ retrievals"

_Atmospheric Chemistry and Physics, 2018_

## Referee Comment (RC1) · Anonymous Referee #1 · 31 Dec 2018

General comments. The authors make contribution to an under-explored topic of understanding differences between the global fluxes estimates based on GOSAT and OCO-2 satellite observations of atmospheric carbon dioxide. By applying same inverse modeling system, same prior fluxes and inverse modeling setup, and using retrievals made with a very similar algorithm, they constructed a good base for comparing performance of the GOSAT and OCO-2 data in application to a problem of quantifying regional carbon fluxes. The study has a potential to contribute to an important problem of understanding the global carbon cycle response to 2015 El-Nino climate anomaly, by providing the alternative views to the phenomenon from the 3 independent observing systems. Although most of important work is already done in this study, more analysis and possibly more extra model runs are required to arrive at robust conclusions, as there is still inconsistency between global annual terrestrial flux estimates made with different observing systems, that should be addressed and elaborated. The study should benefit from giving authors extra time for making necessary revisions.

Specific comments.

Line 1 Title can be simplified to "Terrestrial ecosystem carbon fluxes estimated using GOSAT and OCO-2 retrievals."

Line 66 OCO-2 observations have lower random noise compared to GOSAT, but it is not related to vertical difference in sensitivity. Generally, SWIR observations by both sensors have flat sensitivity to CO2 from the surface to the upper troposphere, thus citing sizable difference between GOSAT and OCO-2 in sensitivity to lower troposphere concentrations requires elaboration, giving more details. Suggest to replace "higher sensitivity near the surface" by "higher sensitivity to column CO2".

Line 171 It is useful to elaborate on application of scaling factors to carbon flux – are those factors applied to total carbon flux in each grid or separately by each optimized component.

Line 237-240 The differences between net flux for 2015 should be related to different atmospheric CO2 growth rate between ground based (also used used in CT2016), GOSAT and OCO-2 observations. Suggest to add those atmospheric CO2 growth rate estimates to comparison, along with inversion-optimized posterior growth rate for all experiments. If the difference between observed ground-based, GOSAT and OCO-2 growth rates is not as much as appears in the inversion results, the inversion setup should be adjusted to provide sufficient constraint on fluxes, so that the growth rate in the inversion optimized simulation matches the observed growth rate.

Technical corrections

Line 26 Suggest to spell out CT2016 as Carbontracker 2016.

[Figure]

Line 43 Need to add "et al." to Chevallier 2007

Line 54 Note that in Takagi et al 2011 only flux uncertainty and uncertainty reduction are estimated, not the fluxes themselves.

Line 82 Reference to GEOS-Chem adjoint is needed here.

Line 105 Replace "Before used" to "Before being used" or "Before using"

Line 212 Add period in Single et al, 2011

Line 219 Uncertainty is assigned to ocean flux, while it was stated that only terrestrial fluxes are not optimized on Line 209-210. Need to make the text consistent.

Line 256 Change 'priori' to 'prior' here and further in the text.

Line 275 Reference is missing.

Line 523 Suggest correcting "Bron" to "Breon"

---

## Referee Comment (RC2) · Anonymous Referee #2 · 2 Jan 2019

Review of "Differences of the inverted terrestrial ecosystem carbon flux between using GOSAT and OCO-2 XCO2 retrievals" by Wang et al

This manuscript presents the results from a numerical experiment in which two inverse estimates of land carbon uptake were made, driven by two different satellite retrievals. One of them results from OCO-2 spectra, while the other one comes from GOSAT. The inverse system is based on GEOS-CHEM and a 4D-VAR method, and spans the year 2015 completely. Posterior fluxes are evaluated by comparing against CT2016 fluxes, and against a set of flask observations, as well as TCCON XCO2 retrievals. The authors conclude that the inversion brings fluxes in closer agreement with all three of

these, and differences between the two flux estimates are discussed in the text. Overall, the manuscript is easy to read and organized logically, and sufficient information is presented to allow the reader to appreciate the results.

What is missing from the current manuscript mostly is scientific depth. The experiment conducted is relatively straightforward, and the text at many points falls into long repetitions of numbers presented already in figures and tables. The differences are highlighted, but what drives these differences, what they imply for the use of these satellite data, and what to learn from the comparisons remains unclear. This does not invalidate the substantial effort, but it brings into question whether a publication like this should be considered scientific literature, or a technical report. I will leave this for the editor to judge.

But even for a technical report, I find the manuscript as presented currently incomplete. The demonstration of smaller biases relative to TCCON and flask observations, and the incidental agreement with CT2016, or GCP, or a set of Asian inversions, brings me to hypothesize that the improvements are not due to the use of the spatially explicit satellite data, but simply a manifestation of a better global total land sink compared to the prior. This can be tested using the poor-man's-inversion first described by Chevallier et al., (2010), in which a global residual land sink (for example that from GCP) is projected onto the land biosphere following the pattern of Net Primary Production. This benchmark is more difficult to beat than a prior from CT2016, as it inherently is globally unbiased and follows patterns of vegetation activity. Improvements beyond those in a poor-man's-inversion due to the use of satellite data would imply that spatial patterns can indeed be estimated from such satellite data, and thus make this manuscript worth reading. Finally, the use of CT2016 as benchmark for a non-satellite inversion seems illogical to me, and should be replaced by a flask-only inversion using the same system as used for the other inversions.

Without these two additions, I feel that this manuscript is not ready for publication in ACP, either as a technical report or as a scientific paper.

[Figure]

A long list of further remarks, and points that require further explanation and discussion comes inside the annotated PDF that accompanies this review.

Please also note the supplement to this comment:
https://www.atmos-chem-phys-discuss.net/acp-2018-1175/acp-2018-1175-RC2-supplement.pdf

**Supplement:**

[revised manuscript text omitted]

---

## Author Comment (AC1) · 25 Apr 2019

**Referee #1:**

We would like to thank the anonymous referee for his/her comprehensive review and valuable suggestions. These suggestions help us to present our results more clearly. In response, we have made changes according to the referee's suggestions and replied to all comments point by point. All the page and line number for corrections are referred to the revised manuscript, while the page and line number from original reviews are kept intact.

**Referee:** General comments. The authors make contribution to an under-explored topic of understanding differences between the global fluxes estimates based on GOSAT and OCO-2 satellite observations of atmospheric carbon dioxide. By applying same inverse modeling system, same prior fluxes and inverse modeling setup, and using retrievals made with a very similar algorithm, they constructed a good base for comparing performance of the GOSAT and OCO-2 data in application to a problem of quantifying regional carbon fluxes. The study has a potential to contribute to an important problem of understanding the global carbon cycle response to 2015 El-Nino climate anomaly, by providing the alternative views to the phenomenon from the 3 independent observing systems. Although most of important work is already done in this study, more analysis and possibly more extra model runs are required to arrive at robust conclusions, as there is still inconsistency between global annual terrestrial flux estimates made with different observing systems, that should be addressed and elaborated. The study should benefit from giving authors extra time for making necessary revisions.

**Response:** We appreciate the referee's suggestion on conducting more model runs and addressing the inconsistency between global annual terrestrial flux estimates from different observing systems. During this revision, we have run two more inversions based on the comments from another referee. In the revised manuscript, in order to make the comparison between satellites based inversions and in situ observations based inversion more consistent, we remove the comparison with CT2016 results, and conduct an inversion using in situ measurements so that we don't have to deal with the differences in transport models and inversion settings between this study and CT2016,

which might compound the comparisons. We also conduct a poor man's inversion to a benchmark (Chevalier et al., 2009), against which more robust evaluation of posterior flux from other three inversions can be made.

To investigate the inconsistency between global annual terrestrial flux estimates from different observing systems, we add evaluations for the satellite XCO2 retrievals using TCCON retrievals and simulated $CO_2$ field as a reference. We found that even with bias-correction applied, both GOSAT and OCO-2 XCO2 retrievals still have relatively large biases. The distinctive differences between GOSAT and OCO-2 data could result in quite different annual global terrestrial land flux estimates.

We have updated the manuscript with new experiments results and more detailed analysis in the revised paper.

**Specific comments**.

Line 1: Title can be simplified to "Terrestrial ecosystem carbon fluxes estimated using GOSAT and OCO-2 retrievals."

**Response:** Thanks for this suggestion. The title has been simplified to the suggested one. See lines 1-2 in the revised manuscript.

Line 66: OCO-2 observations have lower random noise compared to GOSAT, but it is not related to vertical difference in sensitivity. Generally, SWIR observations by both sensors have flat sensitivity to $CO_2$ from the surface to the upper troposphere, thus citing sizable difference between GOSAT and OCO-2 in sensitivity to lower troposphere concentrations requires elaboration, giving more details. Suggest to replace "higher sensitivity near the surface" by "higher sensitivity to column CO2".

**Response:** Many thanks for this suggestion. "higher sensitivity near the surface" has been replaced by "higher sensitivity to column $CO_2$" as suggested. See line 66 of the revised manuscript.

Line 171: It is useful to elaborate on application of scaling factors to carbon flux – are those factors applied to total carbon flux in each grid or separately by each optimized

component.

**Response:** We have added "The scaling factors are applied to each carbon flux components to be optimized monthly in each model grid point." in the first paragraph of Section 2.3.3. See lines 176-177 of the revised manuscript.

Line 237-240:    The differences between net flux for 2015 should be related to different atmospheric CO2 growth rate between ground based (also used used in CT2016), GOSAT and OCO-2 observations. Suggest to add those atmospheric CO2 growth rate estimates to comparison, along with inversion-optimized posterior growth rate for all experiments. If the difference between observed ground-based, GOSAT and OCO-2 growth rates is not as much as appears in the inversion results, the inversion setup should be adjusted to provide sufficient constraint on fluxes, so that the growth rate in the inversion optimized simulation matches the observed growth rate.

**Response:** The differences of net flux are indeed related to the different atmospheric $CO_2$ growth rate between different observing system. However, as shown by Figure 1 in the manuscript, satellite retrievals are unevenly distributed spatially and temporally, which makes it difficult to estimate global atmospheric $CO_2$ growth rate accurately. We have evaluated the uncertainties of the two satellite retrievals using TCCON retrievals, which might better explain the differences of inverted net flux.

For details, please refer to Lines 379 to 391, page 20 in the revised manuscript.

**Technical corrections**

Line 26:    Suggest to spell out CT2016 as Carbontracker 2016.

**Response:** As suggested by another referee, to make the comparisons between different observing systems more consistent, we have replaced CT2016 results with estimates from in-situ observations using our inversion method.

Line 43:    Need to add "et al." to Chevallier 2007

**Response:** We have added "et al." to that reference. See line 44 in the revised manuscript.

Line 54:    Note that in Takagi et al 2011 only flux uncertainty and uncertainty reduction are estimated, not the fluxes themselves.

**Response:** Thanks. We have removed "Takagi et al., 2011" from the reference. See line 55 in the revised manuscript.

Line 82:    Reference to GEOS-Chem adjoint is needed here.

**Response:** The reference "Henze et al., 2007" has been added. See line 82 in the revised manuscript.

Line 105:    Replace "Before used" to "Before being used" or "Before using"

**Response:** We have changed "Before used" to "Before being used". See lines 106-107, page 5 in the revised manuscript.

Line 212:    Add period in Single et al, 2011

**Response:** We have done the revision. See line 215 of the revised manuscript

Line 219:    Uncertainty is assigned to ocean flux, while it was stated that only terrestrial fluxes are not optimized on

**Response:** Actually, ocean flux has also been optimized in our inversions. We have added "Both terrestrial ecosystem $CO_2$ exchanges and ocean flux are optimized in our inversions" in Lines 212-213 of the revised manuscript.

Line 209-210:    Need to make the text consistent.

**Response:** Yes, we have made it consistent. we have added "Both terrestrial ecosystem $CO_2$ exchanges and ocean flux are optimized in our inversions" in Lines 212-213 of the revised manuscript.

Line 256:    Change 'priori' to 'prior' here and further in the text.

**Response:**    Thanks for pointing out this error. We have checked the manuscript and changed all inappropriate "priori" to "prior".

Line 275:    Reference is missing.

**Response:** The missing reference "Gurney et al., 2002" has been added. See line 294, page 14 in the revised manuscript.

Line 523:    Suggest correcting "Bron" to "Breon"

**Response:** We have corrected "Bron" to "Breon". See line 540, page 27 in the revised manuscript.

Reference:

Chevallier, F., et al. (2010), CO2 surface fluxes at grid point scale estimated from a global 21 year reanalysis of atmospheric measurements, J. Geophys. Res., 115, D21307, doi:10.1029/2010JD013887.

---

## Author Comment (AC2) · 25 Apr 2019

**Referee #2:**

We would like to thank the anonymous referee #2 for his/her comprehensive review and valuable suggestions. These suggestions help us to present our results more clearly. In response, we have made changes according to the referee's suggestions and replied to all comments point by point. All the page and line number for corrections are referred to the revised manuscript, while the page and line number from original reviews are kept intact.

**Referee:** This manuscript presents the results from a numerical experiment in which two inverse estimates of land carbon uptake were made, driven by two different satellite retrievals. One of them results from OCO-2 spectra, while the other one comes from GOSAT. The inverse system is based on GEOS-CHEM and a 4D-VAR method, and spans the year 2015 completely. Posterior fluxes are evaluated by comparing against CT2016 fluxes, and against a set of flask observations, as well as TCCON XCO2 retrievals. The authors conclude that the inversion brings fluxes in closer agreement with all three of these, and differences between the two flux estimates are discussed in the text. Overall, the manuscript is easy to read and organized logically, and sufficient information is presented to allow the reader to appreciate the results.

What is missing from the current manuscript mostly is scientific depth. The experiment conducted is relatively straightforward, and the text at many points falls into long repetitions of numbers presented already in figures and tables. The differences are highlighted, but what drives these differences, what they imply for the use of these satellite data, and what to learn from the comparisons remains unclear. This does not invalidate the substantial effort, but it brings into question whether a publication like this should be considered scientific literature, or a technical report. I will leave this for the editor to judge.

But even for a technical report, I find the manuscript as presented currently incomplete. The demonstration of smaller biases relative to TCCON and flask observations, and the incidental agreement with CT2016, or GCP, or a set of Asian

inversions, brings me to hypothesize that the improvements are not due to the use of the spatially explicit satellite data, but simply a manifestation of a better global total land sink compared to the prior. This can be tested using the poor-man's-inversion first described by Chevallier et al., (2010), in which a global residual land sink (for example that from GCP) is projected onto the land biosphere following the pattern of Net Primary Production. This benchmark is more difficult to beat than a prior from CT2016, as it inherently is globally unbiased and follows patterns of vegetation activity. Improvements beyond those in a poor-man's-inversion due to the use of satellite data would imply that spatial patterns can indeed be estimated from such satellite data, and thus make this manuscript worth reading. Finally, the use of CT2016 as benchmark for a non-satellite inversion seems illogical to me, and should be replaced by a flask-only inversion using the same system as used for the other inversions.

Without these two additions, I feel that this manuscript is not ready for publication in ACP, either as a technical report or as a scientific paper. A long list of further remarks, and points that require further explanation and discussion comes inside the annotated PDF that accompanies this review.

**Response:** We accept the referee's comments on the lack of in depth analysis and discussions of the two satellite based inversions in the older manuscript. We are grateful to the referee for suggesting the addition of one inversion using surface observations and another benchmark inversion using global $CO_2$ trend as baseline, which are instrumental in improving our manuscript to its current level. During the revision, we add two inversions as suggested. An evaluation for the two satellite $XCO_2$ retrievals against TCCON $XCO_2$ retrievals and an examination of mismatches from both GOSAT and OCO-2 inversions are performed to better understand the uncertainties of the two satellite retrievals. We add more discussions on probable reasons for the differences of inverted carbon fluxes between the two satellite-based inversions. We have updated the manuscript with new inversion results, rewritten the conclusions and made the necessary modifications accordingly.

**Main changes in the revised manuscripts**: In Section 3, we add subsection 3.1

Inversions using satellite $XCO_2$ retrievals (see Lines 226-232) to describe two satellite-based inversions; subsection 3.2 Inversion using in situ measurements (see Lines 233-242) to describe in situ inversion setting; subsection 3.3 Benchmark inversion (see Lines 243-259) to describe poor man's inversion setting.

In section 4, we rewrite the whole subsection 4.1 (see Lines 262-273) to discuss the global carbon flux updated with results from two added inversions. We also update Table 1 (see Lines 274-276) with results from in situ and benchmark inversions.

In section 4.2, we rephrase the comparison between inverted regional carbon flux from OCO-2 and GOSAT inversions (see Lines 292-313); we added the comparisons of inverted regional flux between all four inversions (see Lines 320-327); we reorganize the discussion on the relationship between the changes of fluxes and satellite data amount for GOSAT and OCO-2 inversions (see Lines 328-353); we redraw the Figure 4 (see Lines 313-315) to add results from in situ and benchmark inversions; we update Table 2 (see Lines 317-318) with results from two new inversions; we add analysis of model-data mismatches from two satellite data based inversions and the discussion on the difference of mismatches being the possible reason for the differences in the inverted flux (see Lines 363-377); we add an evaluation of satellite retrievals against TCCON retrievals and discussions for the impact of uncertainties of satellite retrievals on the inversions (see Lines 378-391); we add statistics of model-data mismatches from GOSAT and OCO-2 inversions into Table 3 (see Lines 355-358); and we add Table 4 (see Lines 392-393), which presents the comparison results of satellite retrievals against TCCON retrievals.

In section 4.3.1 Flask observations (see Lines 393-430), with the addition of the two new inversions and different flask observation dataset adopted, to present new evaluation results, we rewrite this whole subsection to show the uncertainties of posterior mixing ratios from 4 inversions and to discuss the improvement of posterior flux by different inversions; we replace Figure 6 and 8 in the old manuscript with Table 6 (see Lines 464-466) to present the evaluation results using both flask observations and TCCON retrievals; we rewrite the most part of subsection 4.3.2 TCCON

observations (see Lines 441-464) to update the evaluation results and give a more detailed discussion on the improvement of posterior $XCO_2$ by four inversions than the old version of manuscript; we redraw the Figure 9 in the old version of manuscript and rename it as Figure 7 (see Lines 466-467) to present the biases of posterior $XCO_2$ from four inversions at 13 TCCON sites.

In Section 5 Summary and Conclusions (see Lines 471-499): based on the results and analysis, we rewrite this whole section to summarize our findings and give our revised conclusions.

Line 1: This title is not very accessible to a wider public, who likely will not know what an "inverted terrestrial ecosystem carbon flux" is.

**Response:** Thank you for this comment. As suggested by referee #1, we have changed the title to "Terrestrial ecosystem carbon fluxes estimated using GOSAT and OCO-2 $XCO_2$ retrievals."

Line 11: Please mention right away the retrieval versions used, as this has a huge impact on inversion results.

**Response:** We have added the retrieval versions of "Version b7.3" (see Line 13 and 105)

Line 20-21: Given that the total is larger, this would be quite obvious

**Response:** A stronger total global land sink from GOSAT inversion doesn't necessarily means that carbon sink of most regions are stronger than carbon sink of corresponding regions estimated from OCO-2 inversion. For instance, several regions with very large carbon sink might dominate the global total, even with weak carbon sink or weak source in several regions, we may still have a very large global total sink. So this sentence "Regionally, in most regions, the land sinks inferred from GOSAT data are also stronger than those from OCO-2 data." is meaning and we keep it in the manuscript (see Lines 22-23).

Line 24: increased and decreased relative to the prior fluxes, which makes it important to know what these are to appreciate the sentence

**Response:** This sentence is not well organized and causes ambiguity. We have revised this sentence to **"**In temperate regions, the prior land sinks are significantly increased, while in tropical regions the prior land sinks are decreased." (see Lines 25-26)

Line 24: it is hard to see how a conclusion on carbon fluxes can be drawn based on atmospheric $CO_2$ columns and flasks. To believe the qualification "improved" I would need to see an independent comparison to fluxes. Also here, the term "improved" asks for a statement on what the baseline was that you compared to

**Response:** We agree with the referee that it is hard to draw conclusion on flux based $CO_2$ observations. It is an indirect way to evaluate the quality of inverted flux. However, with the lack of direct measurements of flux, there are not much options. Therefore, we follow referee's suggestions to run in situ and benchmark inversions and the comparison with these two inversions can help us to evaluate the quality of inverted flux in satellite-based inversions. We have revised the last sentence of the abstract to "Evaluations using flask and TCCON observations and the comparisons with in situ and benchmark inversions suggest that GOSAT data, can effectively improve the carbon flux estimates in the northern hemisphere." (see Lines 29-31)

Line 41: It is important to stress that these studies were all theoretical, and made many assumptions on the errors and their structure to show the potential of satellite observations. Studies done since then using actual retrieved XCO2 have not confirmed the idea that fluxes can be improved (yet)

**Response:** Thanks for referee for pointing out this. We have revised the sentence to "Studies have shown that, theoretically, satellite observations, though with lower precision than in-situ measurements, can improve the carbon flux estimates". (see Lines 42-43)

Line 45: This description is incorrect: these sensors can sense light at wavelengths affected by atmospheric mixing ratios of $CO_2$, but they never measure concentrations.

**Response:** Yes, you are right. We have replaced "measure" with "retrieve" (see Line 46).

Line 64: I would not say that the community is looking for consensus. Efforts are underway to determine the true fluxes through more direct measurements at the surface.

**Response:** We agree with the referee's comment. We have revised that sentence to "efforts to improve the accuracy of Europe carbon sink estimate are still ongoing". (see Lines 64-65)

Line 75: I am not sure that this is a question of interest to the community. The qualification of which one is better will not drive ongoing work on satellite-based inversions, it is the availability of different datasets over different time periods. In the end, we will use anything that helps us understand the carbon cycle better, and we will discard everything that does not.

**Response:** We agree with the referee that the availability of different datasets over different time periods is important. Nonetheless, before we put all those datasets into one inversion system, we need to know the merits and demerits of every dataset relative to each other. For instance, as added in the revised manuscript, compared with the GOSAT retrievals, OCO-2 retrievals exhibit a relative large positive bias and consequently, give rise to weak posterior land sink. Thus, when we perform the synergy use of two satellite retrievals to constraint carbon flux, the large bias of one satellite retrievals will degrade the improvement by another satellite retrievals with small biases.

Line 81: Abstract said July 2014, please check

**Response:** We have made the correction in the abstract. It is "Oct, 2014" (see Line 81).

Line 81: to = through

**Response:** We have changed "to" to "through" (see Line 81).

Line 83-84: Delete "We analyze the differences of inverted terrestrial ecosystem carbon flux between using two XCO$_2$ data."

**Response:** We have deleted that sentence**.** (see Line 83)

Line 119: Please note here the near lack of any data >60N, which would cover the Boreal regions that are later discussed

**Response:** We have added sentence "Due to the cloud contamination, there are few retrievals in a large portion of tropical land. In northern high latitude area, especially in boreal regions, due to the low soar zenith angle, available satellite retrievals are very sparse." into Section 2.1 to describe the lack of data in northern high latitude." (see Lines 122-124)

Line 131: This suggests many more were available but you decided to limit it to 47? Why??

**Response:** Thank you for this question. Using "chosen" might cause confusion here. Actually, at the time of doing experiment, we downloaded all available dataset from World Data Center for Greenhouse Gases (WDCGG) website. We picked out all flask sites. There were only 47 flask sites with available observations over the year of 2015. Now that we added inversion using in-situ data from Obspack, we replaced flask observations from WDCGG with those from Obspack. We got 56 flask sites with valid observations for 2015. However, there are 4 sites, namely, HUN, HPB, SGP, and TAP, where the standard deviations of first guess mismatch are greater than 5 ppm. So we wouldn't use those 4 sites for evaluation and end up with 52 flask sites. We have updated the relevant text on using flask observations in the manuscript. (see Lines 130-138, 395-405)

Line 143: Is this part of the products you downloaded? If so, then there is no need to mention it here. If not, how was the bias correction done by your team?

**Response:** Yes, it is part of the products we downloaded. In addition, this sentence doesn't provide much information and is repetition of statement made elsewhere. We have deleted that sentence. (see Lines 150-151)

Line 173: Is this term part of the summation?? Please add extra brackets if not

**Response:** It is not part of the summation. We have added extra brackets in the

equation. $J(c) = \frac{1}{2}\sum_{i=1}^{N}(XCO_{2,i}^m - XCO_{2,i}^{obs})S_{obs,i}^{-1}(XCO_{2,i}^m - XCO_{2,i}^{obs}) + (\frac{1}{2}(c - c_a)S_c^{-1}(c - c_a))$ (see Line 179)

Line 186-187: These come from the ACOS data product I presume?

**Response:** Yes. We have added the sentence "These last four quantities are provided from ACOS Version 7.3 Level 2 Lite products." (see Lines 193-194) to give the origin of products.

Line 193, 200,202,203,205,206: I am confused about the fossil fuel fluxes you used. There is a reference to CDIAC here, but it also says that you used the CT2016 fluxes, which according to their website are not the same as CDIAC. Finally, you mention a number of other sources of FF-CO₂ from shipping, airplanes, and oxidation. Your tables however simply quote the CT2016 totals it seems. So what was done??

**Response:** Yes, the fossil fuel fluxes from CT2016 is not the same as CDIAC, but the average of CDIAC and ODIAC products. We have made the correction in the revised manuscript (Lines 202-205). GEOS-Chem $CO_2$ emission module added shipping emission since that part is not included in the CIDAC product. Aviation and oxidation are still part of total fossil fuel emission from CIDAC. GEOS-Chem just deducts from the CIDAC the part of aviation and oxidation and spread them into the level they should be released. Thus the total fossil fuel emission is still kept the same as prescribed.

Line 209: A citation or other proof is needed. I actually have not so much faith in

these fluxes compared to more pCO2 driven estimates such as from Rodenbeck.

**Response:** Yes, it is inappropriate to state that optimized ocean flux from CT2016 represents a more realistic state of ocean flux. However, in this study, first, we focus on the inversion of terrestrial carbon flux. Second, for all four inversion experiments, we use the same ocean flux. Therefore, the choice of ocean flux should not have much impact on the issues we are investigating in this study. We have rewritten the description of the prior flux in the manuscript. (see Lines 210-211)

Line 214: What are the number of degrees of freedom in the matrices? Likely to be less than a few hundred

**Response:** For one month, the number of degrees of freedom is around 470.

Line 217: Please justify this choice, and the difference between lat and lon scales. To me these scales seem very long given the large data density inherent to a satellite inversion. Why can this not be smaller? What would be the effect on the inversion?

**Response:** The difference between latitude and longitude are just for convenience of applying off-diagonal covariance matrix, which is difficult to provide in the GEOS-Chem adjoint model efficiently (Single et al., 2011). The scale length of 500 km is similar to what Chevallier et al. (2010) has used in their study. Though satellite retrievals have better spatial coverage than in-situ measurements, they are not dense enough and not evenly distributed. For instance, there are a few thousand retrievals per month in temperate regions but very sparse retrievals in northern high latitude and tropical regions for assimilation. The longer scale length should extend constraints over regions which are not well sampled by satellites.

Line 220-222: I do not understand this sentence. What are the five and three numbers quoted?

**Response:** The three and five are errors of scaling factors for land and ocean flux respectively. We have corrected that sentence as "the uncertainty of scaling factor for the prior land and ocean fluxes in each month at the grid cell level are assigned to 3

and 5," (see Lines 224-225).

Line 225: I do not see how this procedure accounts for correlated errors in the grid-based retrievals?

**Response:** Satellite $XCO_2$ retrievals are pixel values at instrument viewing resolution and are not grid-based retrievals. Within the model cell box with the size of 2.5x2 degree, there are many pixels which are correlated to some extent. The averaging of those pixels should account for correlated errors.

Line 231: Why did you not also perform an inversion against the surface network data only, to complement the satellite based effort? Now you need to compare your fluxes against CT2016 all the time, when you could have simply created your own. Then you also would not have had to deal with the difference in transport models that now confounds the comparison.

**Response:** We have performed an inversion using surface network data and need not to compare against CT2016. A subsection is added to describe in situ inversion setting in the revised manuscript. (see Lines 233-242)

Line 254: The CT2016 website suggests that this number should be -2.62 PgC/yr. What creates the difference?

**Response:** We have noticed the difference between what we got and the number from CT2016 website. We checked our calculation very carefully. Since we run GEOS-Chem model at a resolution of 2.5x2 degree, we regridded the CT2016 carbon flux at 1x1 degree resolution into our model resolution. The global totals are computed from carbon fluxes at 2.5x2 degree resolution. The regridding procedure we used might cause the difference of our number from CT2016 value. Since we have removed the CT2016 results from our comparison in the revised manuscript (see Lines 275-276), we don't need recheck the calculation again.

Line 254: The CT2016 website has this as 5.51 PgC/yr it seems...?

**Response:** The same reason as mentioned above. (see Lines 275-276)

Line 254: It is difficult to compare atmospheric growth rates directly with flux conversions, as the assumption of instantaneous mixing is that one makes is likely to be false at such short time scales. Please mention this in the text, because now the number of 6.23 looks "better" while it is simply a different metric (that happens to be close to the XCO2 based estimates).

**Response:** Now that we have conducted benchmark inversion, we can use benchmark result as a standard and have removed GCP estimate in the revised paper. (see Lines 275-276)

Line 275: Lack references

**Response:** We have added the missing reference "Gurney et al., 2002". (see Line 294)

Line 278-321: Starting from here, the text becomes a very long description of the numbers that are summarized already in the Table. The words written here do not add anything to add, and the numbers provided are simply copies of the information that is displayed. It can be removed fully from line 278 through line 321.

**Response:** We agree with referee that the description is rather long. However, detailed comparison of regional flux is important for us to understand the impact of satellite retrievals on inversions. Just checking the values in those tables might not be enough. So we will keep the description of comparison and have made this long description more concise. (see Lines 292-313)

Line 322: delete "basically"

**Response:** We have deleted it. (see Line 336)

Line 324: This sounds like an interesting observation: changes in fluxes are largest there where the coverage is least?!

**Response:** What we want to point out is that the changes in fluxes are largest there

where the coverage is the best. The region with least coverage is Northern Boreal Land where the inverted flux has the smallest change relative to the prior flux. (see Lines 335-338)

Line 333: changed from the prior you mean, or do you mean different between two inversions?

**Response:** We mean more carbon flux is changed relative to the prior. We have added "relative to the prior flux" to that sentence. (see Line 347)

Line 340: This sounds quite speculative, yet the statement is also quite obvious. The challenge is to quantify, through your experiments, what the sensitivity to sensor accuracy and spatial resolution is.

**Response:** We agree with the referee that this statement is speculative and obvious. However, based the results we have from our experiments, we are unable to draw any conclusions on the sensitivity to sensor accuracy and the spatial resolution of those two sensors. Thus we have deleted this statement in the revised manuscript. (see Lines 352-353)

Line 343: delete "basically"

**Response:** We have removed the paragraph containing "basically" as pointed out in the first response "**Main changes in the revised manuscripts**".

Line344: delete "CT2016"

**Response:** We have removed the paragraph containing "CT2016" in the section presenting comparison between inversions as pointed out in the paragraphs of "**Main changes in the revised manuscripts**" in the first response.

Line 350: In my opinion this debate is not so intense lately, and the community overall does not believe the large sink to be realistic. And instead of simply remarking on it, your framework could be used to shed some more light on the situation. This would require a flask-based inversion to compare the two satellite-based ones again though.

**Response:** We have performed an in-situ observation-based inversion to compare with our results using satellite retrievals. The carbon sink in Europe estimated from in-situ inversion is close to the estimates from the two satellite inversions. We are still working on the discrepancy of Europe carbon estimate between our in-situ inversion and other in-situ based inversions. In the manuscript, since we have removed CT2016 results related comparison. Thus, most part of this section is rewritten as listed in "**Main changes in the revised manuscripts**".

Line 364: probably better to use "suggest", as there is no conclusion possible based on this evidence discussed.

**Response:** The paragraph using "indicate" has been rewritten.

Line 389: What is quite interesting to discuss is the difference in global land uptake: the total would suggest quite different global growth rates in atmospheric $CO_2$, a metric that we know quite well.

**Response:** We have added more discussions on the difference of different global land uptake estimate. (see Lines 260-276)

Line 408: The fact that these do not improve suggests that it is mostly the new global balance that is improved in the inversions. Hence my request for a benchmark against a poor man's inversion

**Response:** After we conducted in situ inversion and benchmark inversion, we redid the evaluation using flask observations from ObsPack product and found that the little improvement in standard deviations are mainly due to the use of 4 sites, namely, HPA, HUN, SGP and TAP, since our transport model is unable to capture the $CO_2$ levels and variations of these 4 sites. At these 4 sites, the standard deviations of mismatch are larger than 5 ppm. So we exclude these 4 sites from our evaluation. The updated result show improvement in the reduction of both bias and standard deviation. In addition, the addition of benchmark inversion did allow us to better evaluate our satellite data based inversions. We have updated the relevant text in the revised manuscript (see Lines 395-410).

Line 419: For a bias, this is quite large.

**Response:** Yes, we agree with the referee that it is a quite large bias. In the updated manuscript, we have revised the statement and pointed out that the limited improvement is made by OCO-2 retrievals. (see Lines 436-466)

Line 422-423: I am not sure if the evidence agrees with the conclusion presented. First of all, the evidence concerns mixing ratio biases while the conclusion is on fluxes. Second, the reduction of the bias could just as easily have been obtained in the simple "null experiment" that I proposed earlier: take the global growth rate, subtract 2 PgC/yr of ocean fluxes, and spread the remaining flux across the vegetated land weighted by NPP. That land flux transported should be the baseline to beat for an inversion (not the prior flux). See Chevallier et al., (2010, JGR) for more details.

**Response:** We can evaluate the quality of inverted carbon flux indirectly by comparing the posterior $CO_2$ mixing ratio against observations. The reduction of bias of simulated mixing ratio with posterior flux should be a prerequisite for an inversion. We agree with the referee that it is not enough to simply use mixing ratio biases to draw conclusion on the improvement of posterior flux. Therefore, we have performed the poor-man's inversion as a benchmark to evaluate our satellite retrievals-based inversions.

Line 454: It seems that in OCO-2, also the pattern of the bias is conserved and thus little spatial information on the fluxes was derived.

**Response:** Yes, the biases of simulated $XCO_2$ from OCO-2 inversion remain relatively large. As analyzed and discussed in the revised manuscript, there are relatively large positive biases between OCO-2 and TCCON retrievals, and quite small mismatches between prior simulated and OCO-2 $XCO_2$, thus consequently leading to small adjustments to the prior flux. (see Lines 415-430, 436-466)

Line 458-459:I think you mean the other way around Change the sentence to ",

using the GEOS-Chem 4D-Var data assimilation 472-473)system."(472-473)

**Response:** We have changed the sentence to "using the GEOS-Chem 4D-Var data assimilation system." (see Lines 472-473)

Line 464:This is confusing. The land net flux is a positive number if one includes fossil fuels and biomass burning, and now it becomes larger in the inversions? I think you described the opposite in the text: the land sink increased from the prior and thus a smaller net land flux when all components are included.

**Response:** It should be "excluding fossil fuels and biomass burning". We have replaced including with excluding. (see Line 478)

Line 467-468:    This is not a good representation of the data: the growth rate is a measured quantity in the atmosphere, and should be compared to the simulated mixing ratios at a set of background sites of one wants to comment on its realism. For the global total fluxes, one can compare them to GCP which is also a flux estimate. But please do not refer to a GCP global growth rate and mention it to be actually fluxes, this is incorrect.

**Response:** We now benchmark two satellite data based inversion with the poor man's inversion and no longer use GCP estimate as a standard. We have removed GCP related statement in the revised manuscript.

Line 470: delete "basically"

**Response:** We have deleted it. (see Line 485)

Line 470: Based on what metric do you qualify this as significant? Is the change outside the uncertainty specified? If the significance was not evaluated, then do not use this qualification.

**Response:** Use "significantly", we just tended to emphasize the large adjustment of priori flux made at northern and southern temperate regions in GOSAT inversion.

This wording may cause confusion. So we change it to "largely". (see Line 485)

Line 480: "carbon fluxes of OCO-2 and GOSAT": this does not exist, these instruments measure radiances. Please rephrase more accurately.

**Response:** We have revised to "carbon fluxes estimated from OCO-2 and GOSAT $XCO_2$ retrievals". (see Line 493)

Line 480: Again, based on what metric?

**Response:** With added inversion experiments, we can compare the simulated $CO_2$ field with posterior flux from four inversions against surface observations. The reduction of biases and standard deviations at global and regional scale could be taken as metric to do the evaluation. In the revised paper, we have added more detailed analysis and discussion of those statistics. (see Lines 406-430, 436-463)

Line 480: Again, I would bet that the poor man's inversion also causes such an improvement and it does not rely on these instruments at all...

**Response:** The poor man's inversion does cause similar improvement in reducing bias. However, there is litter improvement in the standard deviation of mismatch. As shown in the revised manuscript, GOSAT inversion shows evident improvement over benchmark. (see Line 406-430, 436-463)

Line 484: This statement seems not true for GOSAT, for which biases seem to be equally large or even larger after the inversion.

**Response:** This statement is not clearly and may denotes improvement in Southern Hemisphere. Actually, we had hoped to point out the increase of biases by GOSAT inversion. In southern hemisphere, for GOSAT, the biases are indeed equally large or even larger after the inversion. We have corrected that sentence "biases are elevated to a certain extent to" to "biases are slightly increased". (Line 493)

Reference:
Chevallier, F., Feng, L., Bösch, H., Palmer, P. I., and Rayner, P. J.: On the impact of

transport model errors for the estimation of $CO_2$ surface fluxes from GOSAT observations, Geophys. Res. Lett., 37, L21803, https://doi.org/10.1029/2010GL044652, 2010.

Singh, K., Jardak, M., Sandu, A., Bowman, K., Lee, M., and Jones, D.: Construction of non-diagonal background error covariance matrices for global chemical data assimilation, Geosci. Model Dev., 4, 299-316, https://doi.org/10.5194/gmd-4-299-2011, 2011.

---

## Author Response (AR2)

Dear Editor,

According to the reviewers' comments and suggestions, we have made major revision to our manuscript. The main changes in the manuscript are as follows:

1) As suggested by one reviewer, we reorganized the section 4, we present the evaluation results first and then followed by the flux analysis and discussions.

2) For better clarity, we rewritten the abstract, we removed the statement on the differences between posterior fluxes from satellites and prior fluxes and focus more on the comparisons between satellites and in situ inversion.

3) We removed the comparisons between satellites inversions and poor-man inversion on regional carbon flux, but added more analysis about the comparisons between the satellites and in situ inversions.

4) We changed the statemen of "benchmark inversion" to "poor-man inversion", and made clear the purpose and the calculations of doing poor-man inversion.

5) We made the conclusion clear that GOSAT data can effectively improve the carbon flux estimates in Northern Hemisphere and its performance is close to in situ data, while OCO-2 data, with the specific version used in this study, shows only slight improvement.

6) We also checked errors and typos carefully and made the necessary corrections.

The point-by-point response to the reviews and the detailed changes are listed in the attachments. Many thanks to you and the referees for the time and effort you expend on this paper.

Best Regards,

Sincerely yours,

Fei Jiang

**Referee #1:**

We thank the anonymous referee for his/her valuable comments and constructive suggestions. We have made changes according to the referee's suggestions and replied to all comments point by point. All the page and line number for corrections are referred to the revised manuscript, while the page and line number from original reviews are kept intact.

**Referee:** General comments.

The aim of the study is to provide a comparison of simulated $CO_2$ concentrations estimated with inverse model using OCO-2 and GOSAT retrievals for the year 2015. The questions of interest to broader audience are differences in the amplitudes of respective flux corrections and their spatial distributions, as well as how well the optimized simulations agree with the observed concentrations by surface flask and TCCON networks. Authors discuss the spatial variability of the satellite data biases with respect to TCCON data and the biases in inverse model estimated concentrations with respect to surface flask and TCCON data. Based on comparison of the estimated fluxes to benchmark inversion and posterior fits to ground-based observational data not used in inversion, authors conclude that use of GOSAT data in inversion results in better fit to observations than OCO-2 data. The manuscript is been resubmitted and was substantially revised with respect to earlier version. Presentation of the material is clear and has improved over initial submission, so manuscript can be accepted after minor revisions. The text should also be checked for orthographical errors.

**Response:** We appreciate the referee's insightful comments. We checked for orthographical errors carefully and made the necessary corrections.

Detailed comments

L74 Authors consider if "current OCO-2 observations have a greater potential than GOSAT …". It would be useful to note what would be the reasons affecting usefulness of OCO-2 or GOSAT? Did they mean spatially and temporally varying biases?

**Response:** Yes, spatially and temporally varying biases affect usefulness of satellite retrievals greatly. As pointed out by Chevalier et al. (2007), biases of a few tenth of one ppm in $XCO_2$ could bias subcontinental flux estimates by several tenths of gigaton of carbon. Spatial coverage also affects the usefulness of satellite data, especially in regions with frequent clouds, where OCO-2 is anticipated to perform better. In the revised manuscript, we have changed that sentence to "it is still not clear whether with the improved monitoring capabilities and better spatial coverage, current OCO-2 observations have a greater potential than GOSAT observations for estimating $CO_2$ flux at regional or finer scale, since except spatial coverage, the biases also affect the usefulness of satellite retrievals greatly.". See lines 77-80, pages 3-4.

L454 Although results of the analysis support the conclusions, it should be noted that the study period of 2015 is characterized by strong El-Nino and the spatial distribution of fluxes typical for more common non El-Nino years, appears disturbed in the El-Nino year.

**Response:** Thanks for the referee for calling our attentions to the influence of El-Nino event. We understand the importance of evaluating performance of two satellite data in both El-Nino and non El-Nino years. However, the availability of overlap of OCO-2 and GOSAT data from ACOS for only 20 month prevents us from doing multi-year inversions.

L456 The uncertainty/bias in TCCON retrievals is cited, but evidence for the bias is not shown/discussed. Suggest to add some reference(s) on TCCON biases.

**Response:** TCCON retrievals are subject to air-mass dependent and air-mass independent biases. The correction factors are applied to the column-averaged mole fractions. The air-mass dependent correction factor is determined from the symmetric component of the diurnal variation. The air-mass-independent correction factor is determined by comparisons with in situ profiles measured over TCCON sites from aircraft or balloon payloads. The TCCON biases are usually evaluated by aircraft or balloon profile observations. However, the comprehensive evaluation of TCCON biases are still hindered from the lack of enough profile data. We have added two

references of Wunch et al. (2010) and Messerschmidt et al. (2011) in the revised manuscript in line 336, page 15.

L490 Analysis of the posterior fit to surface flask observations in Southern and North hemispheres indicates there are biases in GOSAT and OCO-2, and those are changing in different directions. Comparison to TCCON show the retrieval bias difference between GOSAT and OCO-2 are in order of 1 ppm. It is worth noting that, while observed mean difference with TCCON calls for correction of the GOSAT and OCO-2 data, based on mean deviation from TCCON, it was not done in this study, as opposed to some other inverse modeling studies.

**Response:** The bias difference up to 1 ppm between GOSAT and OCO-2 retrievals against TCCON retrievals does seem rather large. However, due to limited number of collocated satellite retrievals, the sample size for computing bias is relatively small. Therefore, the large bias difference should be treated as a relative value. As shown in Table 4, when comparing to the same prior $CO_2$ mixing ratios, the difference of overall mismatches between GOSAT and OCO-2 data is 0.57 ppm, suggesting the bias difference might not be as large as shown by comparison with TCCON data. As described in Section 2, GOSAT and OCO-2 retrievals used in our inversions are already bias-corrected. The remaining biases of satellite retrievals suggest that the bias-correction scheme implemented need to be improved. However, due to the short time period of our inversions, the sparseness of TCCON sites and the lack of profile data, it is really difficult to figure out an appropriate way to further reduce satellite retrievals bias. The following sentences have been added in the revised manuscript to point out the deficiency of statistics, see lines 487-493, page 25.

"…It should be noted that due to the limited number of collocated satellite retrievals, the real bias difference might not be up to 1 ppm. As shown in Table 4, the difference of overall mismatches between GOSAT and OCO-2 data is 0.57 ppm. These indicate that although both OCO-2 and GO-SAT products were bias-corrected using TCCON retrievals, the uncertainties of OCO-2 and GO-SAT retrievals are still very large, especially for OCO-2 retrieval, resulting the worse performance of OCO-2 retrieval, which also suggest that the bias-correction scheme implemented may need to be improved."

Suggested technical corrections

L68 Suggest to change "constrain the surface carbon flux inversion" to "constrain the surface carbon fluxes"

**Response:** We have changed "constrain the surface carbon flux inversion" to "constrain the surface carbon fluxes". See lines 70-71, page 3.

L250 Fprior mistyped

**Response:** We have corrected "Fpiror" to "Fprior" in the revised manuscript. Seen line 259, page 12.

L256 suggest changing "by multiply by" to "by multiplying by"

**Response:** We have changed "by multiply by" to "multiplying by" in the revised manuscript. See lines 268-269, page 12.

L276 'In situ' to 'in situ'

**Response:** We have change "In situ" to "in situ". See line 363, page 17

L377 abbreviated TCCON station names (such as 'Bial') should be explained somewhere in the text.

**Response:** We have added the full name of TCCON stations in section 2.2 in the revised manuscript. See lines 149-151, page 7.

L392 in "0.34 to0.59" space is missing

**Response:** We have added space. See line 485, page 25.

L408 in "0.93ppm" space is missing

**Response:** We have added space. See line 286, page 13.

L415 Instead of Figure 7, Figure 6 should be referred here, as comparisons to TCCON is on Line 447.

**Response:** We have corrected "Figure 7" to "Figure 4" since according to another referee's comments, the section 4 has been reorganized in the revised manuscript, and Figure 6 is renamed to Figure 4. See lines 345-346, page 16.

**Referee #2:**

We thank the anonymous referee for his/her valuable comments and constructive suggestions. We have made changes according to the referee's suggestions and replied to all comments point by point. All the page and line number for corrections are referred to the revised manuscript, while the page and line number from original reviews are kept intact.

**Referee:** I was pleased to receive the response of the authors to my comments, and I appreciate the extra effort that was done to address my concerns. I think the manuscript has improved substantially in this revision, but not yet enough for publication in ACP. The manuscript needs quite a few small modifications of errors, typos, and mistakes in at least one figure. It can also profit from rewriting and reordering parts of the text. And before publication can proceed, the authors need to consider the terms of use of the ObsPack data that they downloaded, as currently they did not comply with them. Please find below my additional comments on the new manuscript presented.

**Response:** We are very grateful to the referee's insightful comments and really appreciate his/her patience while we were working on the manuscript. In the revised manuscript, we have reorganized the section 4. We present the evaluation results first and then followed by the flux analysis. For better clarity, we rewrite the abstract, part of section 4 and conclusions as well. We corrected the mistake in the Figure 4. We also checked errors and typos carefully and made the necessary corrections. We will explain how we follow the terms of use of the ObsPack data in detail.

Main comments:

The use of the ObsPack instead of the flasks is in principle a good idea. I do not understand why the authors decided to download a carbontracker obspack though, as the website that explains this product (https://www.esrl.noaa.gov/gmd/ccgg/obspack/release_notes.html#obspack_co2_1_C ARBONTRACKER) explicitly suggests not to use these files as primary source of data for inversions, but instead to get the latest real obspack from the website. It also reminds the user explicitly to comply with the terms of use of these data in a study, which means that (1) all data providers need to be contacted before publication to explain the use of their data and to agree on the way to acknowledge them, (2) a citation to the dataset through its DOI must be included in the text. I repeated this Fair Use Statement in my comments below. Without complying with these rules, the current manuscript should not be published.

**Response:** Many Thanks for this comment and suggestion. The reason for using CarbonTracker Obspack other than the full ObsPack data is just for convenience. Before we decided to use CarbonTracker ObsPack data, we did read the release note carefully and understand the suggestions for not using those data as primary source for inversions. The latest real ObsPack data contain much more measurements than those used by CarbonTacker. We don't have much experience with assimilating in situ measurements. It is a lot of works to filter out the measurements not suitable for assimilation. It is also not an easy task for us to figure out the appropriate observation uncertainties. Therefore, in order to finish the in situ inversion and complete the revision of the manuscript on time, we chose to use CarbonTracker Obspack data to do the inversion.

We did follow the terms of use of OpsPack data closely. After we downloaded the ObsPack data, we read the terms of use carefully and emailed to all ObsPack PIs to acknowledge the use of data and inquire the proper citation of ObsPack data. We got replies from Dr. Andres Schmidt and Dr. Andy Jacobson and they agreed to let us use the data. All the other PIs didn't reply to us and so we assumed no objections from them for using the data. A copy of email we sent and the replies from Dr. Andres Schmidt and Dr. Andy Jacobson are listed as follows.

(1) Email we sent to all data providers on Apr 7, 2019:

**acknowledgement of using obspack_co2_1_CARBONTRACKER_CT2016_2017-02-06 data**

发件人：wanghm<wanghm@nju.edu.cn>
时 间：2019年4月7日(星期天) 凌晨0:23
收件人：bev.law;Casper.Labuschagne<Casper.Labuschagne@weathersa.co.za>;
h.a.scheeren<h.a.scheeren@rug.nl>;swofsy<swofsy@seas.harvard.edu>;vandenbulk<vandenbulk@ecn.nl>;
colm.sweeney<colm.sweeney@noaa.gov>;leuenberger<leuenberger@climate.unibe.ch>;
wpaplawsky<wpaplawsky@ucsd.edu>;john.mund<john.mund@noaa.gov>;haszpra.l<haszpra.l@met.hu>;
pshepson<pshepson@purdue.edu>;lukasz.chmura<lukasz.chmura@fis.agh.edu.pl>;necki<necki@agh.edu.pl>;
Gordon.Brailsford<Gordon.Brailsford@niwa.co.nz>;vandinther<vandinther@ecn.nl>;
frank.meinhardt<frank.meinhardt@uba.de>;josep-anton.morgui<josep-anton.morgui@ic3.cat>;
Paul.Steele<Paul.Steele@csiro.au>;anna.karion<anna.karion@noaa.gov>;
marc.delmotte<marc.delmotte@lsce.ipsl.fr>;huilin.chen<huilin.chen@rug.nl>;e.gloor<e.gloor@leeds.ac.uk>;
scpiper<scpiper@ucsd.edu>;Doug.Worthy<Doug.Worthy@ec.gc.ca>;kenneth.c.aikin<kenneth.c.aikin@noaa.gov>;
arlyn.andrews<arlyn.andrews@noaa.gov>;pieter.tans<pieter.tans@noaa.gov>;jeff.peischl<jeff.peischl@noaa.gov>;
Paul.Krummel<Paul.Krummel@csiro.au>;francesco.apadula<francesco.apadula@rse-web.it>;
accox<accox@ucsd.edu>;lvgatti<lvgatti@ipen.br>;Ray.Langenfelds<Ray.Langenfelds@csiro.au>;
ingrid.vanderlaan;am12721<am12721@bristol.ac.uk>;
Marcel.vanderSchoot<Marcel.vanderSchoot@csiro.au>;petri.keronen<petri.keronen@helsinki.fi>;
roger.curcoll<roger.curcoll@ic3.cat>;sjwalker<sjwalker@ucsd.edu>;zimnoch<zimnoch@agh.edu.pl>;
olli.peltola<olli.peltola@helsinki.fi>;meelis.molder<meelis.molder@nateko.lu.se>;
s.odoherty<s.odoherty@bristol.ac.uk>;MSTorn<MSTorn@lbl.gov>;SCBiraud<SCBiraud@lbl.gov>;
Michel.Ramonet<Michel.Ramonet@lsce.ipsl.fr>;Andres.Schmidt<Andres.Schmidt@oregonstate.edu>;
abollenbacher<abollenbacher@ucsd.edu>;mon<mon@m.tohoku.ac.jp>;ghg_obs<ghg_obs@met.kishou.go.jp>;
janne.levula<janne.levula@helsinki.fi>;Michal.Heliasz<Michal.Heliasz@cec.lu.se>;
Eric.J.Hintsa<Eric.J.Hintsa@noaa.gov>;h.a.j.meijer<h.a.j.meijer@rug.nl>;
michal.galkowski<michal.galkowski@agh.edu.pl>;kenneth.schuldt<kenneth.schuldt@noaa.gov>;
ingeborg.levin;tuula.aalto<tuula.aalto@fmi.fi>;clm<clm@nilu.no>;
thomas.b.ryerson<thomas.b.ryerson@noaa.gov>;oh<oh@nilu.no>;
samuel.hammer<samuel.hammer@iup.uni-heidelberg.de>;hensen<hensen@ecn.nl>;
ed.dlugokencky<ed.dlugokencky@noaa.gov>;frumau<frumau@ecn.nl>;andrew.uea<andrew.uea@gmail.com>;
marcus.schumacher<marcus.schumacher@dwd.de>;kirk.w.thoning<kirk.w.thoning@noaa.gov>;
kathryn.mckain<kathryn.mckain@noaa.gov>;MLFischer<MLFischer@lbl.gov>;
Sylvia.Nichol<Sylvia.Nichol@niwa.co.nz>;g.forster<g.forster@uea.ac.uk>;fred.moore<fred.moore@noaa.gov>;
aoki<aoki@m.tohoku.ac.jp>;alex.vermeulen<alex.vermeulen@nateko.lu.se>;
martin.steinbacher<martin.steinbacher@empa.ch>;stephens<stephens@ucar.edu>;
pasi.kolari<pasi.kolari@helsinki.fi>;nakazawa<nakazawa@m.tohoku.ac.jp>;agomezp<agomezp@aemet.es>;
James.W.Elkins<James.W.Elkins@noaa.gov>;safshar<safshar@ucsd.edu>;rkeeling<rkeeling@ucsd.edu>;
john.b.miller<john.b.miller@noaa.gov>;j.turnbull<j.turnbull@gns.cri.nz>;juha.hatakka<juha.hatakka@fmi.fi>;
yniwa<yniwa@mri-jma.go.jp>;ysawa<ysawa@mri-jma.go.jp>;tmachida<tmachida@nies.go.jp>;
hmatsued<hmatsued@mri-jma.go.jp>;andy.jacobson<andy.jacobson@noaa.gov>

Dear Obspack data providers:

We are preparing a manuscript already posted as discussion paper in ACP, tilted " Differences of the inverted terrestrial ecosystem carbon flux between using GOSAT and OCO-2 XCO2 retrievals"(acp-2018-1175). It is currently under the review stage. Upon the request of reviewers, we need to use in-situ data from "obspack_co2_1_CARBONTRACKER_CT2016_2017-02-06" dataset to do the inversion of surface carbon flux and also to evaluate our inversion results from satellite retrievals. Since it is our first time to use OBSPACK data, please let us know how we can follow the data use policy properly. We would like to know how to cite this dataset since the citation is not sent to us. Do we need to provide reference for every dataset? If so, would data providers please kindly send us the reference of your dataset? If there is any concern on our use of this dataset, please just let us know. Many thanks!

Best regards,

Hengmao Wang
Associate Professor
International Institute of Earth System Science
Nanjing University (2) The reply from Dr. Andy Jacobson on Apr 7, 2019

**Re: acknowledgement of using obspack_co2_1_CARBONTRACKER_CT2016_2017-02-06 data**

发件人：Andy Jacobson<andy.jacobson@noaa.gov>
时 间：2019年4月7日(星期天) 凌晨1:22
收件人：wanghm<wanghm@nju.edu.cn>; bev.law;
Casper.Labuschagne<Casper.Labuschagne@weathersa.co.za>; h.a.scheeren<h.a.scheeren@rug.nl>;
swofsy<swofsy@seas.harvard.edu>; vandenbulk<vandenbulk@ecn.nl>; colm.sweeney<colm.sweeney@noaa.gov>;
leuenberger<leuenberger@climate.unibe.ch>; wpaplawsky<wpaplawsky@ucsd.edu>;
john.mund<john.mund@noaa.gov>; haszpra.l<haszpra.l@met.hu>; pshepson<pshepson@purdue.edu>;
lukasz.chmura<lukasz.chmura@fis.agh.edu.pl>; necki<necki@agh.edu.pl>;
Gordon.Brailsford<Gordon.Brailsford@niwa.co.nz>; vandinther<vandinther@ecn.nl>;
frank.meinhardt<frank.meinhardt@uba.de>; josep-anton.morgui<josep-anton.morgui@ic3.cat>;
Paul.Steele<Paul.Steele@csiro.au>; anna.karion<anna.karion@noaa.gov>;
marc.delmotte<marc.delmotte@lsce.ipsl.fr>; huilin.chen<huilin.chen@rug.nl>; e.gloor<e.gloor@leeds.ac.uk>;
scpiper<scpiper@ucsd.edu>; Doug.Worthy<Doug.Worthy@ec.gc.ca>; kenneth.c.aikin<kenneth.c.aikin@noaa.gov>;
arlyn.andrews<arlyn.andrews@noaa.gov>; pieter.tans<pieter.tans@noaa.gov>; jeff.peischl<jeff.peischl@noaa.gov>;
Paul.Krummel<Paul.Krummel@csiro.au>; francesco.apadula<francesco.apadula@rse-web.it>;
accox<accox@ucsd.edu>; lvgatti<lvgatti@ipen.br>; Ray.Langenfelds<Ray.Langenfelds@csiro.au>;
ingrid.vanderlaan; am12721<am12721@bristol.ac.uk>;
Marcel.vanderSchoot<Marcel.vanderSchoot@csiro.au>; petri.keronen<petri.keronen@helsinki.fi>;
roger.curcoll<roger.curcoll@ic3.cat>; sjwalker<sjwalker@ucsd.edu>; zimnoch<zimnoch@agh.edu.pl>;
olli.peltola<olli.peltola@helsinki.fi>; meelis.molder<meelis.molder@nateko.lu.se>;
s.odoherty<s.odoherty@bristol.ac.uk>; MSTorn<MSTorn@lbl.gov>; SCBiraud<SCBiraud@lbl.gov>;
Michel.Ramonet<Michel.Ramonet@lsce.ipsl.fr>; Andres.Schmidt<Andres.Schmidt@oregonstate.edu>;
abollenbacher<abollenbacher@ucsd.edu>; mon<mon@m.tohoku.ac.jp>; ghg_obs<ghg_obs@met.kishou.go.jp>;
janne.levula<janne.levula@helsinki.fi>; Michal.Heliasz<Michal.Heliasz@cec.lu.se>;
Eric.J.Hintsa<Eric.J.Hintsa@noaa.gov>; h.a.j.meijer<h.a.j.meijer@rug.nl>;
michal.galkowski<michal.galkowski@agh.edu.pl>; kenneth.schuldt<kenneth.schuldt@noaa.gov>;
ingeborg.levin; tuula.aalto<tuula.aalto@fmi.fi>; clm<clm@nilu.no>;
thomas.b.ryerson<thomas.b.ryerson@noaa.gov>; oh<oh@nilu.no>;
samuel.hammer<samuel.hammer@iup.uni-heidelberg.de>; hensen<hensen@ecn.nl>;
ed.dlugokencky<ed.dlugokencky@noaa.gov>; frumau<frumau@ecn.nl>; andrew.uea<andrew.uea@gmail.com>;
marcus.schumacher<marcus.schumacher@dwd.de>; kirk.w.thoning<kirk.w.thoning@noaa.gov>;
kathryn.mckain<kathryn.mckain@noaa.gov>; MLFischer<MLFischer@lbl.gov>;
Sylvia.Nichol<Sylvia.Nichol@niwa.co.nz>; g.forster<g.forster@uea.ac.uk>; fred.moore<fred.moore@noaa.gov>;
aoki<aoki@m.tohoku.ac.jp>; alex.vermeulen<alex.vermeulen@nateko.lu.se>;
martin.steinbacher<martin.steinbacher@empa.ch>; stephens<stephens@ucar.edu>;
pasi.kolari<pasi.kolari@helsinki.fi>; nakazawa<nakazawa@m.tohoku.ac.jp>; agomezp<agomezp@aemet.es>;
James.W.Elkins<James.W.Elkins@noaa.gov>; safshar<safshar@ucsd.edu>; rkeeling<rkeeling@ucsd.edu>;
john.b.miller<john.b.miller@noaa.gov>; j.turnbull<j.turnbull@gns.cri.nz>; juha.hatakka<juha.hatakka@fmi.fi>;
yniwa<yniwa@mri-jma.go.jp>; ysawa<ysawa@mri-jma.go.jp>; tmachida<tmachida@nies.go.jp>;
hmatsued<hmatsued@mri-jma.go.jp>

Dear Dr Wang,

Apologies to all for cross-posting.

You can read about the usage terms for all of our ObsPack products at
https://www.esrl.noaa.gov/gmd/ccgg/obspack/. At that site, you will find
explicit instructions for how to cite the products and how to
acknowledge data providers who have contributed measurements to the
products. If you have further questions along those lines, please send
them to me and Ken Schuldt (kenneth.schuldt@noaa.gov).

One important aspect of the terms of use for ObsPack products is that
you may need to invite data providers to be coauthors on your paper.
This depends on how the measurements are used in your paper, and is a
conversation you need to have with each data provider. Thank you for
starting this process.

The CarbonTracker ObsPack is not the one I would recommend for inversion
studies or evaluation. Instead, the GLOBALVIEW+ CO2 ObsPack, currently
at version 4.2, is the preferred product for finding the measurements you need. You can use a CarbonTracker ObsPack (although you should instead use the more recent CT2017 one), but that is an indirect way of gaining access to the measurements. GLOBALVIEW+, on the other hand, is produced specifically for the purposes you cite.

I have read your manuscript and I do not need to be a coauthor on it. I agree that your OCO-2 and GOSAT results will be easier to interpret if you also have an inversion using in situ CO2 measurements using your system. You should definitely consider comparing your results to the more recent CT2017 instead of CT2016, as our system has been significantly improved. I ask that you follow our usage terms laid out at https://www.esrl.noaa.gov/gmd/ccgg/carbontracker/citation.php, which in this case would involve an edit on line 26 to correctly identify "NOAA's CarbonTracker, version CT2016", and to add our requested acknowledgments text to your acknowledgments section.

Best Regards,

Andy Jacobson

On 4/6/19 10:23 AM, wanghm wrote:
> Dear Obspack data providers:
>

(3) The reply from Dr. Andres Schmidt on Apr 8, 2019

**using obspack_co2_1_CARBONTRACKER_CT2016_2017-02-06 data**

发件人：Andres Schmidt<andres.schmidt.osu@gmail.com>
时 间：2019年4月8日(星期一) 上午10:16
收件人：wanghm<wanghm@nju.edu.cn>
抄 送：andres.schmidt<andres.schmidt@oregonstate.edu>; andres.schmidt<andres.schmidt@geo.rwth-aachen.de>

Dear Hengmao Wang, as far as I am concerned I am happy to see that the data from the carbontracker/obspack sites I am associated with (andres.schmidt@oregonstate.edu)are being used in your manuscript.

Sincerely,

Andres Schmidt
Dept. of Forest Ecosystems & Society
Oregon State University
Cirvallus, OR, USA
(now at RWTH Aachen andres.schmidt@geo.rwth-aachen.de)

We are preparing a manuscript already posted as discussion paper in ACP, tilted " Differences of the inverted terrestrial ecosystem carbon flux between using GOSAT and OCO-2 XCO2 retrievals"(acp-2018-1175). It is currently under the review stage. Upon the request of reviewers, we need to use in-situ data from "obspack_co2_1_CARBONTRACKER_CT2016_2017-02-06" dataset to do the inversion of surface carbon flux and also to evaluate our inversion results from satellite retrievals. Since it is our first time to use OBSPACK data, please let us know how we can follow the data use policy properly. We would like to know how to cite this dataset since the citation is not sent to us. Do we need to provide reference for every dataset? If so, would data providers please kindly send us the reference of your dataset? If there is any concern on our use of this dataset, please just let us know. Many thanks!

Best regards,

Hengmao Wang
Associate Professor
International Institute of Earth System Science
Nanjing University

Although I like the introduction of the poor-man's inversion, the description in the methods section seems incorrect to me, and I find the way it is integrated into the study not very strong. This comes from the choice to use it as an extra inverse solution from the beginning, and to discuss its flux results alongside that of the other inversions. But the poor-man's inversion can only be used to look at the global total flux (which it matches by design), and to look at the distribution of $CO_2$ mixing ratios and XCO2 values across the globe. This it should follow reasonably well, thus setting a benchmark to beat for real inverse solutions. Currently, the label "benchmark" is used throughout the text including that of "benchmark inversion" which is confusing: the flux result of this poor-man's method is the one thing one should *not* put much emphasis on, especially not below the global total scale. It is therefore also no use to show its regional flux solution in Table 2 and in Fig 4, nor be discussed in Section 4.2 in my opinion.

**Response:** Thank you for this comment and suggestion. The poor-man inversion conducted in this study was exactly according to the Chevallier's approach. The description of the method was combined from the descriptions of Chevallier et al. (2009) and Chevallier et al. (2010). The difference between Chevallier's approach and ours is that to be consistent with our three other inversions, we set prior flux uncertainty proportional to prior flux in poor-man inversion, while in Chevallier et al. (2010), it was set proportional to the heterotrophic respiration flux of ORCHIDEE, and in Chevallier et al. (2009), it was set prior flux uncertainty proportional to the gross carbon fluxes.

However, we agree with the referee that the way of integrating poor-man inversion into this study is not strong. In the revised manuscript, we have changed all "benchmark inversion" to "poor-man inversion", removed poor-man inversion results from Table 2 and Fig 4, and removed the comparisons and discussions of regional carbon fluxes of poor-man inversion result in Section 4.2 in the revised manuscript. It should be noted that since section 4 was reorganized, now, Table 2 and Fig 4 are renamed to Table 3 and Fig 6, and Section 4.2 is renamed as Section 4.3 in the revised manuscript.

It is a bit awkward that the reader is first learning a lot about GoSAT to OCO-2 flux differences and how their regional budgets differ in great detail in Section 4.2, but only later in Section 4.3 learns that the OCO-2 inversion is not very trustworthy and is not able to reproduce the atmospheric XCO2 and surface $CO_2$ better than the poor-man's inversion (which can be called a benchmark in this context). So in fact, all I read earlier becomes then in a sense irrelevant. Please consider bringing the assessment of the quality of the inversions forward in the manuscript, so that the flux analysis that comes afterwards can focus more on the relevant part of the study (GoSAT and in-situ inverse results). OCO-2 can then be still discussed, but only to indicate whether GoSAT satellite results are corroborated or not by OCO-2.

**Response:** Thanks for the referee's suggestion. In the revised manuscript, we have reorganized Section 4 and present the assessment of the quality of the inversions first in Section 4.1, and the flux analysis on Global budget and regional fluxes afterward in Section 4.2 and Section 4.3. We also add more analysis about the comparisons between the satellites and in situ inversions as follows, which is shown in lines 409-420, pages 20-21 in the revised manuscript.

"Compared with the in situ inversion, in the boreal regions, the land sinks estimated from GOSAT and OCO-2 inversions are much weaker than those from in situ inversion, especially in the Eurasian Boreal, the land sink estimated by in situ inversion is more than two times larger than the estimates of GOSAT and OCO-2 inversions. In the tropical land, the total land sinks inferred from both GOSAT and OCO-2 inversions are weaker than those from the in situ inversion, but in different regions, the situations are different. In the Temperate lands, except for Europe and south Africa, the land sinks from GOSAT and OCO-2 inversions are much stronger than those from the in situ inversion. For example, in South America Temperate, GOSAT inversion shows a strong carbon sink, while in situ inversion shows a weak source. For different continents, in North America, Asia, Europe, the carbon sinks inferred from GOSAT inversion are comparable to those from in situ inversion, while in South America and Africa, the carbon sinks inferred from OCO-2 inversion are much closer to the in situ inversions."

Abstract: I think that the text does not summarize so well the main findings anymore, and should be rewritten. The main message should focus on the posterior fluxes compared to the in-situ inversion, and not comparing the two satellites to the prior. Then, one can highlight that the main difference on the largest scale is the latitudinal distribution of land sinks, with the satellites suggesting a smaller Boreal and Tropical sink, combined with larger temperate sinks in both the NH and SH. However, OCO-2 and GoSAT generally do not agree on which continent contains the smaller or larger sinks. Also, the comparison of the simulated surface mixing ratios and XCO2 columns shows that only GoSAT and the in-situ inversion perform better than a poor-man's solution that closes the annual global mass balance of $CO_2$. This puts the usefulness of the OCO-2 retrieval product used here into question.

**Response:** Many thanks for this suggestion. We have rewritten the abstract. In the revised manuscript, we removed the statement on the differences between posterior fluxes from satellites and prior fluxes and focus more on the comparisons between satellites and in situ inversion. We highlight the following conclusions:

(1) the terrestrial ecosystem carbon sink (excluding biomass burning emissions) estimated from GOSAT data is stronger than that inferred from OCO-2 data and weaker than the in situ inversion, and matches the poor-man inversion to be the best.

(2) Regionally, in most regions, the land sinks inferred from GOSAT data are also stronger than those from OCO-2 data, and in North America, Asia, Europe, the carbon sinks inferred from GOSAT inversion are comparable to those from in situ inversion. For the latitudinal distribution of land sinks, the satellites-based inversions suggest a smaller bo-real and tropical sink, but larger temperate sinks in both Northern and Southern Hemispheres than the in situ inversion. However, OCO-2 and GOSAT generally do not agree on which continent contains the smaller or larger sinks.

(3) Evaluations using flask and TCCON observations and the comparisons with in situ and poor-man inversions suggest that only GOSAT and the in situ inversions perform better than a poor-man's solution. GOSAT data can effectively improve the carbon flux estimates in Northern Hemisphere, while OCO-2 data, with the specific version used in this study, shows only slight improvement.

For details, please refer to lines 15-17, lines 20-22, lines 24-32, pages 1-2.

List of remarks:

page 1, line 15 "benchmark inversion": I would refer to the latter as a poor-man's inversion in which only the global $CO_2$ growth rate is projected onto the land biosphere, to be used as a benchmark for the simulated atmospheric $CO_2$ distributions of the real inversions.

**Response:** We have changed "benchmark inversion" to "poor-man inversion" and rephrase the sentence as "One inversion for the comparison, using in situ $CO_2$ observations, and another inversion as a benchmark for the simulated atmospheric $CO_2$ distributions of the real inversions, using global atmospheric $CO_2$ trend and referred as poor-man inversion, are also conducted."
For details, please refer to lines 15-17, page 1.

page 1, line 22: "more consistent with …" simply say that the GoSAT-based inversion seems to best capture the observed global $CO_2$ growth rate.

**Response:** We have rephrased that sentence as "…estimated from GOSAT data is stronger than that inferred from OCO-2 data, weaker than the in situ inversion, and matches the poor man inversion to be the best." See lines 21-22, page 1.

Page 2, line 29: it is worth to say explicitly that the OCO-2 retrieval you used here seems unfit for inverse modeling, but that later versions seem to perform better (Chevallier et al., 2019, ACPD). I also urge the authors to focus their future efforts on the later retrieval products from OCO-2.

**Response:** We have added one sentence "OCO-2 data, with the specific version used in this study, show only slight improvement" (see lines 31-32, page 2) to point out the poor performance of OCO-2 product used in this study. We also mention in the end of conclusion section that the improved performance of newer version of OCO-2 product. "... It also should be noted that though the OCO-2 $XCO_2$ retrievals of version b7.3 used in this study perform worse than GOSAT data and in situ measurements in our inversions, one recent study has shown that the newer version of OCO-2 data has a much better performance in constraining carbon flux (Cheval-lier et al., 2019). With constantly improved retrieval algorithm and bias-correction scheme, more robust estimate of carbon flux from satellite XCO2 retrievals could be achieved."

For details, see lines 31-32, page 2, and lines 528-533, page 26.

Page 4, line 84: please do not use "benchmark inversion" to label this flux product, but explain the purpose of this approach better.

**Response:** We have changed all "benchmark inversion" to "poor-man inversion" and given more explanation on the purpose of using poor-man inversion.

"For comparisons, one inversion based on in situ measurements is conducted, and another simple one, which uses the global $CO_2$ trend as a benchmark for the simulated atmospheric $CO_2$ distributions of the real inversion, is also implemented."

See lines 86-88, page 4.

Page 6, line 128: This is where my main comment comes into play. The Fair Use Statement given in the readme file of the Obspack you downloaded was:

**ObsPack Fair Use Statement**

#

**This cooperative data product is made freely available to the scientific community and is intended to stimulate and support carbon cycle modeling studies. We rely on the ethics and integrity of the user to assure that each contributing national and university laboratory receives fair credit for their work. Fair credit will depend on the nature of the work and the requirements of the institutions involved.**

**Your use of this data product implies an agreement to contact each contributing laboratory for data sets used to discuss the nature of the work and the appropriate level of acknowledgement. If this product is essential to the work, or if an important result or conclusion depends on this product, co-authorship may be appropriate. This should be discussed with the appropriate data providers at an early stage in the work. Contacting the data providers is not optional; if you use this data product, you must contact the applicable data providers. To help you meet your obligation, the data product includes an e-mail distribution list of all data providers.**

\# This data product must be obtained directly from the ObsPack Data Portal at www.esrl.noaa.gov/gmd/ccgg/obspack/ and may not be re-distributed. In addition to the conditions of fair use as stated above, users must also include the ObsPack product citation in any publication or presentation using the product. The required citation is included in every data product and in the automated e-mail sent to the user during product download.

**Response:** As answered in the major comments part, we paid close attention to the fair use of data and followed the terms of use of data as required.

Page 7, line 147: insert "area" between shaded and shows

**Response:** We have inserted "area". See line 156, page 8.

Page 9, line 196: This is yet another reference "$CO_2$ trend" to the poor-man's inversion. Pleas try to introduce it better, and use it consistently please.

**Response:** We have rephrased the sentence as follow: "Three inversions, using GOSAT data, OCO-2 data, and in-situ measurements, are conducted from Oct 1, 2014 to December 31, 2015, respectively. Poor-man inversion, based on global atmospheric $CO_2$ trend and using poor-man's method (Chevallier et al, 2009, 2010), is also conducted." in the revised manuscript. See lines 204-207, page 10.

Page 11, line 232: descripted = described

**Response:** We have corrected "descripted" to "described". See Line 242, page 11.

Page 12, line 249: I do not understand this formula and I wonder if a mistake was made. piror = prior (typo). But why do you add something proportional to the prior flux uncertainty, instead of proportional to GPP? And why do you need trial-and-error to determine the scaling factor k? This is not the same approach as taken by Chevallier, whom you cite for this approach.

**Response:** Thank you for this comment.

(1) Yes, "piror" is a typo, we have corrected "piror" to "prior". See line 259, page 12

in the revised manuscript.

(2) The poor-man inversion conducted in this study was exactly according to the Chevallier's approach. In the introduction of this method, we combined the descriptions from Chevallier et al. (2009) and Chevallier et al. (2010).

In the page 4 of Chevallier et al. (2009), the method is described as follows:

"…The ocean fluxes are kept identical, to the prior ones. Over land, the inverted fluxes $x_{pm}$ are defined as

$$x_{pm} = x_b - k\sigma$$

where k is a unique scaling factor and $\sigma$ is the vector made of the prior error standard deviations, i.e., the square root of the diagonal of **B**. Here k was chosen by trial and error so that the mean global total of the $x_{pm}$ fluxes equals the mean global total of fluxes inverted from the surface measurements over the 3-year period. A value of 1/55 was found. This simple approach aims at matching the mean global growth rate of CO2, which is too large with our prior fluxes over land (see the end of section 2.1.2), without any spatial or temporal information from the observations. In practice, it distributes the land carbon sink according to the gross carbon fluxes from the vegetation.".

In the page 8 of Chevallier et al. (2010), it was described as follow:

"…In this baseline (which is slightly simplified here), the ocean fluxes are kept identical to the prior ones. Over land the poor man's flux $F$pm at location (*x, y*) and at time *t* is defined as

$$F_{pm}(x, y, t) = F_{prior}(x, y, t) + k\,(year) \times \sigma(x, y, t)$$

$F_{prior}$ *(x, y, t)* is the prior flux at the same time and location. $\sigma(x,y,t)$ is its uncertainty, that is, the standard deviation of the prior error described in section 2.1. *k(year)* is a coefficient that varies as a function of the year only. *k* is chosen here so that the mean annual global totals of the poor man's fluxes equal the mean global totals given by the annual global $CO_2$ growth rate from the *GLOBALVIEW*-$CO_2$ [2009] product multiplied by a conversion factor (2.12 GtC a$^{-1}$ per ppm [Denman et al., 2007, Table 7.1]), In practice, this simple approach distributes the land carbon sink according to the heterotrophic respiration fluxes from the vegetation without any spatial information from the atmospheric observations or any temporal information within any given year.".

In these two papers, $\sigma$ was explicitly defined as prior flux uncertainty. The difference between their approach and ours is that to be consistent with our three other inversions, we set prior flux uncertainty proportional to prior flux in poor-man inversion, while in Chevallier et al. (2010), it was set proportional to the heterotrophic respiration flux of ORCHIDEE, and in Chevallier et al. (2009), it was set prior flux uncertainty proportional to the gross carbon fluxes.

For the calculation of the coefficient of $k$, we agree the referee that we don't need to do trial-and-error to determine it, $k$ can be solved directly from the formula as

$$k = \left(\sum F_{pm} - \sum F_{prior}\right)/\sum \sigma \tag{1}$$

Where $\sum F_{pm}$ equals the global totals given by the observed annual global $CO_2$ growth rate. During the calculation, since on different time scale, the $\sigma$ is different and the global annual uncertainty is not simply the summation of each grid per hour, we calculated several times and got different coefficient of $k$ for different time scale. That is why we said that we did trial-and-error to determine k. However, anyway, we found that whatever did we calculate on monthly or annual time scale, the final $F_{pm}$ distributed on each grid and each three hours are the same. Therefore, the statement of "k is determined by trial-and-error" is indeed improper. We have changed this statement in the revised manuscript. For details, please refer to lines 260-265, page 12.

Chevallier, F., Engelen, R. J., Carouge, C., Conway, T. J., Peylin, P., Pickett‐Heaps, C., Ramonet, M., Rayner, P. J., and Xueref-Remy, I.: AIRS‐based versus flask‐based estimation of carbon surface fluxes, J. Geophys. Res., 114, D20303, doi:10.1029/2009JD012311, 2009.

Chevallier, F., Ciais P., Conway T.J., Aalto T., Anderson B.E., Bousquet P., Brunke E.G., Ciattaglia L., Esaki Y., Fröhlich M., Gomez A., Gomez-Pelaez A.J., Haszpra L., Krummel P.B., Langenfelds R.L., Leuenberger M., Machida T., Maignan F., Matsueda H., Morguí J.A., Mukai H., Nakazawa T., Peylin P., Ramonet M., Rivier L., Sawa Y., Schmidt M., Steele L.P., Vay S.A., Vermeulen A.T., Wofsy S., and Worthy D.: $CO_2$ surface fluxes at grid point scale estimated from a global 21 year reanalysis of atmospheric measurements, J. Geophys. Res., 115, D21307, 2010.

Page 12, line 260: "inverted global carbon budgets" please remove "inverted"

Response: We have removed "inverted". See line 351, page 16.

Page 12, line 263: "benchmark inversion" please rewrite

Response: We have replaced "benchmark inversion" with "poor-man inversion". See line 355, page 17.

Page 16, line 316: In my opinion the benchmark inversion is not very useful here, as its regional flux simply reflects global GPP and not a piece of information derived from the data like in the actual inversions. I suggest to remove it here, and in Fig 4.

Response: Thank you for this suggestion. We have removed results of poor-man inversion in Table 2 and Fig 4 which are now renamed as Table 3 and Fig 6 in the revised manuscript. See lines 404-407, page 20.

Page 17, line 321: "close to the benchmark result": by writing this, you suggest to the reader that it is a good thing for the inversions to be close to the benchmark. But for continental fluxes this is not true at all, and this is why I think this gives the wrong message when put into the figure/text/table.

Response: Thank you for this comment. We have removed the comparison of regional carbon fluxes with "benchmark inversion" in the revised manuscript. Seen lines 409-420, pages 20-21.

Page 20, line 380: Why is this section here, and not part of Section 4.3 where once again a comparison to TCCON is presented? And why are the other two results (in situ and benchmark) not shown? I think it would help to group these results together.

Response: This paragraph only gives comparisons between TCCON $XCO_2$ retrieval and GOSAT and OCO-2 $XCO_2$ retrievals. The aim of these comparisons is to show the uncertainties of OCO-2 and GOSAT retrievals, so as to explain the reason for the different performances of OCO-2 and GOSAT retrievals in the inversions. At the beginning of that paragraph, we have emphasized this objective using the sentences of "Moreover, the uncertainties of OCO-2 and GOSAT retrievals may be another reason for the different performances in these two inversion experiments. We use TCCON retrieval to evaluate the uncertainties of OCO-2 and GOSAT XCO2 retrievals." (See lines 473-475, page 24). We found that although both OCO-2 and GOSAT products were bias-corrected using TCCON retrieval, there are larger mismatches between OCO-2 and TCCON than those between GOSAT and TCCON, and the mismatches among different sites of GOSAT are more consistent than OCO-2, indicating that the uncertainties of OCO-2 products are larger than GOSAT ones, resulting worse performance of OCO-2 retrieval than that of GOSAT retrieval. The objective of the paragraph is different from those in section 4.3.2, which shows the evaluation of posterior $XCO_2$ against TCCON data. Therefore, we still don't combine this paragraph with section 4.3.1 in the revised manuscript. However, in order to make it clear, we reorganized part of that paragraph as follows:

"…At most sites except Garm, OCO-2 retrievals have positive biases, while GOSAT retrievals tend to have negative bias except at Bial and Garm sites. It also could be found that the spread of GOSAT data biases are small, falling in the range of -0.36 to -0.58 ppm at most sites, while the spread of OCO-2 data biases is relatively large, with biases greater than 0.7 ppm at more than half of sites, and in the range of 0.34 to 0.59 ppm only at 3 sites. Overall, GOSAT retrievals (-0.46 ppm) have lower bias than OCO-2 retrievals (0.6 ppm) and the difference between two retrievals is relatively large. It should be noted that due to the limited number of collocated satellite retrievals, the real bias difference might not be up to 1 ppm. As shown in Table 4, the difference of overall mis-matches between GOSAT and OCO-2 data is 0.57 ppm. These indicate that although both OCO-2 and GOSAT products were bias-corrected using TCCON retrievals, the uncertainties of OCO-2 and GOSAT retrievals are still very large, especially for OCO-2 retrieval, resulting the worse performance of OCO-2 retrieval, , which also suggest that the bias-correction scheme implemented may need to be improved."

See lines 481-493, pages 24-25.

Page 21, line 392, space missing and typo in "to0.59 pm"

**Response:** We have added space and corrected the typo. See line 485, page 25.

Page 21, line 409: Please make clear that this is not surprising because part of these evaluation data were used in the inversion in that case

Response: Thank you for this suggestion. We have rewritten the sentence as follows:

"Not surprisingly, in situ inversion, using surface observations which include all the flask measurements used for evaluation, shows the best improvement in posterior $CO_2$ mixing ratio with"

See line 288, page 13.

Page 22, line 426: litter = little

**Response:** We have changed "litter" to "little". See line 306, page 14.

Page 23, line 436: "ground XCO2 observations", please simply write "We use data from 13 TCCON sites to…"

**Response:** We have rewritten the sentence as "We also use data from 13 TCCON sites to…".  See line 317, page 14.

Page 23, line 436: Please make clear that also here the comparison is not fully independent: the TCCON data were used in the bias correction scheme of at least OCO-2 (I don't know about GoSAT but I suspect the same there).

Response: Yes, TCCON data were also used in the bias correction scheme of GOSAT product. We have added a sentence here to point out that the comparison is not fully independent.

 "It should be noted that the comparisons of posterior $XCO_2$ from GOSAT and OCO-2 inversions with TCCON data are not fully independent since the TCCON data were used in the bias-correction scheme of both GOSAT and OCO-2 products (Wunch et al.,

2011)."

See Line 320-323, page 15.

Page 23, line 437: into = onto

**Response:** We have changed "into" to "onto". See line 318, page 14.

Page 24, line 461: The fact that only the in-situ inversion beats the benchmark on all 4 numbers should be mentioned in the text.

Response: Thank you for this suggestion. We have mentioned it as follows, and see lines 339-342, page 15 of the revised manuscript.

"…Overall, it also could be found from Table 1 that only in situ inversion beats the poor-man inversion on all 4 statistics, followed by GOSAT inversion, which beats the poor-man on 3 statistics, indicating that in situ measurements have the best performance in the inversion, and GOSAT retrieval have similar performance as in situ data."

Page 25, Figure 7: There seems to be an error in the figure: the bars for benchmark and in-situ are exactly the same for all sites. Please check and fix this.

**Response:** Thank you very much! Yes, we made an error in this figure. We have redrawn this figure, in the revised manuscript, it has been renamed as Figure 4. See line 348, page 16.

Page 26, line 490: I would not say that OCO-2 could improve the modeling of $CO_2$ concentrations: your poor-man's inversion shows that you can achieve better results by simply scaling your fluxes to match the global growth rate of $CO_2$.

**Response:** We have rephrased the statement as follow:

"Evaluations of the inversions using $CO_2$ concentrations from flask measurements and TCCON retrievals show that the simulated $CO_2$ concentrations with GOSAT posterior fluxes are much closer to the observations than those with OCO-2 estimates…."

For details, see Line 515-523, page 27.

Page 26, line 492: "bench inversion" incorrect

**Response:** We have replaced "bench inversion" with "poor-man inversion". See line 520, page 27.

Page 26, line 495 "GOAST" typo

**Response:** We have corrected "GOAST" to "GOSAT". See line 520, page 27.

[revised manuscript text omitted]

---

## Author Response (AR3)

**Referee #1:**

We thank the anonymous referee #1 for his/her valuable comments and constructive suggestions. We have made changes according to the referee's suggestions and replied to all comments point by point. All the page and line number for corrections are referred to the revised manuscript, while the page and line number from original reviews are kept intact.

**Referee**: General comments

The manuscript presents results of the inverse modeling study, comparing the regional carbon flux estimates made separately with in-situ, OCO-2 and GOSAT observations. Authors found that among two satellite data products, GOSAT-based estimates of regional CO2 fluxes for 2015 appear closer to those made with in-situ data, than ones made with OCO-2 data. The manuscript has been revised after being sent back for major review. Authors properly addressed the review questions and suggestions; thus it can be published with technical corrections on minor issues appearing in the revised text.

Detailed comments

In response to L490 comments by 1st reviewer, authors write 'The bias difference up to 1 ppm between GOSAT and OCO-2 retrievals against TCCON retrievals does seem rather large'. It contradicts with the notice of sizable effect of sub-ppm retrieval biases on fluxes as mentioned by Chevallier et al (2007), cited in response to L74 comment by the authors. This is non critical note as it doesn't affect the text directly.

Response: Thank you for this comment. In response to L490 comments by 1st reviewer, we try to point out that the seemingly large bias differences between GOSAT and OCO-2 retrieval against TCCON observations should be treated as relative values and the real bias differences might not be that large. Thus it is not contradictory to the notice of sizable effect of sub-ppm retrieval biases on fluxes.

Line 535

In the Acknowledgements, it is advisable to mention contribution by Obspack in-situ data providers (rep name/organization, or organization)

Response: We have added the acknowledgements of contributions of ObsPack in-situ data providers and TCCON PIs as well. Since there are more than 30 laboratories involved in the ObsPack product, we don't list the names of those organizations in the Acknowledgements. See Page 27, Line 539-545.

Editorial/technical corrections

Line 79 Suggest revising 'since except spatial coverage, the biases …' to 'since the biases …'

Response: We have revised "since except spatial coverage, the biases" to "since the biases …". See Page 4, Line 79.

Line 108 Replace 'Roggers' with 'Rogers'

Response: We have replaced "Roggers" with "Rogers". See Page 5, Line 108.

Line 483 Revise 'might not be up to 1 ppm' to 'might be below 1 ppm'

Response: We have revised "might not be up to 1 ppm" to "might be below 1 ppm". See Page 25, Line 483.

Line 486 Revise 'resulting the worse performance' to 'resulting in the degraded performance'

Response: We have revised "resulting the worse performance" to "resulting in the degraded performance". See Page 25, Line 486.

[revised manuscript text omitted]